# Traditional Use, Phytochemical Profiles and Pharmacological Properties of *Artemisia* Genus from Central Asia

**DOI:** 10.3390/molecules27165128

**Published:** 2022-08-11

**Authors:** Aliya Nurlybekova, Aidana Kudaibergen, Aizhan Kazymbetova, Magzhan Amangeldi, Aizhamal Baiseitova, Meirambek Ospanov, Haji Akber Aisa, Yang Ye, Mohamed Ali Ibrahim, Janar Jenis

**Affiliations:** 1The Research Center for Medicinal Plants, Al-Farabi Kazakh National University, al-Farabi Ave. 71, Almaty 050040, Kazakhstan; 2Research Institute for Natural Products & Technology, Almaty 050046, Kazakhstan; 3University of Chinese Academy of Sciences, Beijing 100049, China; 4State Key Laboratory of Drug Research, Shanghai Institute of Materia Medica, Chinese Academy of Sciences, Shanghai 201203, China; 5National Center for Natural Products Research, School of Pharmacy, University of Mississippi, Oxford, MS 38677, USA; 6Xinjiang Technical Institutes of Physics and Chemistry, Central Asian of Drug Discovery and Development, Chinese Academy of Sciences, Urumqi 830011, China

**Keywords:** *Artemisia genus*, TKM, Kazakhstan, sequiterpene lactone, essential oils, flavonoids, α-glucosidase, PTP1B, BNA, antioxidant

## Abstract

The flora of Kazakhstan is characterized by its wide variety of different types of medicinal plants, many of which can be used on an industrial scale. The Traditional Kazakh Medicine (TKM) was developed during centuries based on the six elements of ancient Kazakh theory, associating different fields such as pharmacology, anatomy, pathology, immunology and food nursing as well as disease prevention. The endemic *Artemisia* L. species are potential sources of unique and new natural products and new chemical structures, displaying diverse bioactivities and leading to the development of safe and effective phytomedicines against prevailing diseases in Kazakhstan and the Central Asia region. This review provides an overview of *Artemisia* species from Central Asia, particularly traditional uses in folk medicine and the recent numerous phytochemical and pharmacological studies. The review is done by the methods of literature searches in well-known scientific websites (Scifinder and Pubmed) and data collection in university libraries. Furthermore, our aim is to search for promising and potentially active *Artemisia* species candidates, encouraging us to analyze Protein Tyrosine Phosphatase 1B (PTP1B), α-glucosidase and bacterial neuraminidase (BNA) inhibition as well as the antioxidant potentials of *Artemisia* plant extracts, in which endemic species have not been explored for their secondary metabolites and biological activities so far. The main result of the study was that, for the first time, the species *Artemisia scopiformis* Ledeb. *Artemisia albicerata* Krasch., *Artemisia transiliensis* Poljakov, *Artemisia schrenkiana* Ledeb., *Artemisia nitrosa* Weber and *Artemisia albida* Willd. ex Ledeb. due to their special metabolites, showed a high potential for α-glucosidase, PTP1B and BNA inhibition, which is associated with diabetes, obesity and bacterial infections. In addition, we revealed that the methanol extracts of *Artemisia* were a potent source of polyphenolic compounds. The total polyphenolic contents of *Artemisia* extracts were correlated with antioxidant potential and varied according to plant origin, the solvent of extraction and the analytical method used. Consequently, oxidative stress caused by reactive oxygen species (ROS) may be managed by the dietary intake of current *Artemisia* species. The antioxidant potentials of the species *A. schrenkiana*, *A. scopaeformis*, *A. transiliensis* and *Artemisia scoparia* Waldst. & Kitam. were also promising. In conclusion, the examination of details between different *Artemisia* species in our research has shown that plant materials are good as an antioxidant and eznyme inhibitory functional natural source.

## 1. Introduction

The study of Kazakhstan extant sources has shown the territory of Kazakhstan to be rich with a variety of medical skills and ideas that were accumulated and cultivated among the Kazakh people throughout history and carefully preserved by subsequent generations to be part of Kazakh folk medicine [1]. The traditional book “Shipagerlik Bayan” (“Confessions of a Healer”), written by by the Kazakh medical doctor Oteyboydak Tleukabyl in the 15th century AD, explained in detail the medical theories of pharmacology, anatomy, pathology, immunology and food nursing in Traditional Kazakh Medicine (TKM). Each theory represents a system containing six main elements, namely, kengistik tugyr (space), turak tugyr (earth), suwik tugyr (coldness), ystyk tugyr (heat), zharyk tugyr (light) and karangy tugyr (darkness) [2,3,4]. Ancient Kazakh folk medicine arose based on the practical experiences of many generations. Conventionally, all medicinal plants used by Kazakhs were divided into fortifying, refreshing, warming and laxative plants [1,5] and were effectively used against several diseases including bronchitis, bronchial asthma, urethritis, chronic rheumatoid arthritis, stomach pain, high acidity, diarrhea, hemostasis, metrorrhagia, venomous snake bites and cancer [1,6,7].

Ancient TKM was based on the use of medicinal plants as well as tinctures, teas, decoctions and powders that were prepared from them. A number of such medicines included rose, ferruginous, citrus seed, pomegranate, chicory, elecampane and wormwood. [1]. Among the examples of such plant materials widely used in Kazakhstan are plants belonging to the genus *Artemisia* L., which is one of the most widely distributed perennial herbal plants. The name of the genus *Artemisia* comes from the Greek word “Artemis”, which means “healthy” [8]. This well-known medicinal plant genus has been used traditionally in a form of infusions, tinctures and decoctions. Hydro/alcoholic liquid *Artemisia* species extracts improve digestion, stimulate the appetite and are used for dyspepsia, acid gastritis, gastrointestinal tract diseases, liver diseases, the gall bladder, insomnia, malaria, influenza and to treat upper respiratory tract maladies. Additionally, they have been employed in the treatment of bronchial asthma, rheumatism, eczema, dysentery, rheumatism, anemia, jaundice, obesity, meteorism, migraine, hypertension and tuberculosis. Central Asia’s folk medicine uses the infusion of *Artemisia* species flowers for ulcerative colitis, an inflammatory process in the cecum, hemorrhoids, bad breath, epilepsy and several other diseases (Figure 1) [9]. *Artemisia* species has a great economic and medicinal perspective due to the fact of the isolation of artemisinin, which is the main phytochemical of *Artemisia annua* L., is represented as an antimalarial drug. This research encouraged researchers all over the world to study traditional plants with the purpose of finding new bioactive medicines [10].

The most famous *Artemisia* species (wormwood) occupies a special place in the healing of the Kazakh due to its well-established healing properties. There are any relevant examples, such as *Artemisia cina* O.C. Berg et C.F. Schmidt (dermene), which has been used in folk medicine for heart attacks and cancers of the stomach, esophagus and duodenum, where the leaves are collected before the flower opens and then the flowers and stems are harvested. For the treatment of asthma, bronchitis and inflammatory diseases, they drink boiled *A. cina* seeds [11]. Additionally, *A. cina* seeds, in combination with raisins, crushed and mixed, are used to treat lung diseases [12]. The medicinal plant *Artemisia rupestris* L. has been used for gastrointestinal tract and liver diseases, cancer, antidotes and various skin and mucous membranes diseases [13,14]. The use of *Artemisia* species in the world’s traditional medicine is widely documented, demonstrating the genus’s high ethnopharmacological importance. *Artemisia annua*, for example, is mentioned in several ancient books as being useful for the treatment of consumptive fever, jaundice, summer heat wounds, tuberculosis, lice, scabies, dysentery and hemorrhoids, as well as in the form of pain relievers, and in Iran, it is used as an antispasmodic, carminative or sedative remedy for children [15] *Artemisia afra* Jacq. ex Willd., in traditional African pharmacopeia, is used to relieve inflammation and pain for many tribes. Furthermore, its infusion is used in the treatment of malaria [16]. Cervicitis is treated with *Artemisia vulgaris* in Iranian Traditional Medicine [17].

Such a broad variety of phytochemical activities owes to the presence of several active ingredients and secondary metabolites. According to the literature, among 260 *Artemisia* species revealed, various classes of secondary metabolites were shown, including lignans, sesquiterpenoids, flavonoids, coumarins, glycosides, caffeoylquinic acids, sterols and polyacetylenes [18,19,20]. Terpenes, particularly sesquiterpene lactones, which are typical for *Artemisia*, are extremely diverse and plentiful and exhibit a wide range of therapeutic effects: anticancer, antimalarial, anti-inammatory, immunomodulatory, antiulcerogenic, antibacterial, antifungal and antiviral effects [21]. Many species of *Artemisia* genus, such as *A. rupestris*, *Artemisia frigida* Willd., *Artemisia annua* L. and *Artemisia lavandulaefolia* DC., have been described in The Kazakh Herbal Medicine (2012) and are widely used in TKM for the treatments of influenza, liver diseases, diarrhea, wounds, beriberi, tuberculosis, nervous disorders, headaches and toothaches and to regulate pressure and weaken the processes of joint swelling [6].

In Russian folk medicine, *Artemisia* extracts of leaves and roots are used for liver, stomach and spleen diseases. A solution of fresh *Artemisia* juice with dilute ethanol is used for the treatment of kidney stones and insomnia and as an anti-inflammatory agent [22,23]. In Kazakhstan, *Artemisia* tincture (the aboveground part of the plant) is used in folk medicine for stomach ulcers, caecum, hemorrhoids, unpleasant mouth odors and epilepsy [18]. Bitter *Artemisia* is used as a spice for fatty products in the food industry. It is also used for flavoring some alcoholic beverages and wines [24]. Furthermore, the aboveground part is used as an important raw material in the paint industry [25].

The leaves are harvested during the flowering period, the smell of the raw materials is a little balsamic and the taste is slightly bitter. The unlignified parts of the root are harvested in the fall, when the stem dries up completely, or in the spring, before the plant begins to grow back. Herbal medicine is predominantly applied in Asian countries and in many other countries for the treatment of several (mild and severe) ailments and infectious diseases. *A. annua* has been shown as an effective therapy against the malaria pathogen *Plasmodium falciparum* [26,27]. In addition, synthetic drugs commonly used as medicaments can have unpredictable serious side effects and are always associated with a high treatment cost, bringing a serious limitation of therapy. Therefore, the development of new therapeutic strategies is urgently needed, especially ones supported via traditional medicine, which are known as alternative therapeutic methods. Currently, most people around the world are relying on traditional plant medicine to meet their treatment needs, and 60% of the drugs in pharmacies are obtained from medicinal plants [28]. Nowadays, there is a global trend towards the use of medicinal plants and their associated formulations as preventive and therapeutic agents for many diseases because of their availability, safety, potency and valuable traditional knowledge [29,30,31]. Due to the rising demand for phytomedicines on a worldwide scale, the discovery for novel therapeutic plants is essential [32]. Plant-based medications have been employed in several epidemics throughout history. For instance, they were implemented during the two previous outbreaks of coronavirus—MERS-CoV in 2012 and SARS-CoV in 2013. Further, they have been used in the recent seasonal epidemics caused by influenza virus and dengue fever [33,34,35,36]. Moreover, in an emergency situation, such as the presence of COVID-19, and within the absence of therapy, the development of effective or potential synthetic drugs and vaccines is subject to time limits inherent in validation by research protocols and clinical trials.

Herbal medicines and natural products that are readily available and proven to be safe can save you time as your first line of defense. Its antiviral properties have been poorly studied, but the research published so far is promising. Thus, research should be encouraged in the current setting of a race against time to develop a cure for the COVID-19 pandemic [37].

Antioxidants could be used as potential agents for preventing and treating oxidative stress-related disorders [38], including Alzheimer’s disease, diabetes, cancer, arthritis, cardiovascular disease [39] and the ageing process [40]. The most dangerous part of oxidative stress is the formation of reactive oxygen species (ROS) [38,41], which constantly form in the living cell as products of normal oxygen metabolism [42]. ROS spontaneously or in the presence of transition metals converts to more aggressive radicals [43], which can cause damage to many cellular components—for example, lipids, DNA and proteins [44]. Due to the potentially harmful toxicity and carcinogenicity, synthetic antioxidants are not favored for use in the food and drug industries [45]. Therefore, natural ones have become important to use to protect food and enhance human health [46]. Protein Tyrosine Phosphatase 1B (PTP1B, EC 3.1.3.48) and α-glucosidase (EC 3.2.1.20) are the most crucial enzymes for diabetes mellitus, which is a chronic disorder evoked by the high level of blood sugar [47]. PTP1B appears as a key regulator of insulin-receptor activity that acts at the insulin receptor and downstream signaling components, such as the insulin receptor substrate [48]. The α-Glucosidase enzyme is found in the small intestine and catalyzes the breakdown of sugar into glucose [49]. The bacterial neuraminidase BNA (EC 3.2.1.18) is from the group of exo-sialidases, which cleaves the α-ketosidic bond connecting the terminal sialic acid residue with the adjoining oligosaccharide fragment [50]. Sialic acid linkage is very necessary for infections by pathogenic bacteria such as *Clostridium perfrigens* [51].

Here, a wide review of the literature about *Artemisia* species was performed, aiming to provide a short description of the folk uses of the *Artemisia* species in Central Asia, particularly in Kazakhstan, as well as a description of the bioactive components isolated from this important medicinal plant genus. We also encourage the investigation of PTP1B, α-glucosidase and BNA inhibition as well as the antioxidant potentials of *Artemisia* plants, which phytochemical and pharmacological analyses have not reported.

## 2. Distribution of *Artemisia* L. in Central Asia

*Artemisia* genus is one of the largest plant genera belonging to the *Asteraceae* family, with about 500 species distributed all over the world [52]. *Artemisia* species are found across temperate North America, the Mediterranean area, Asia, Africa and Australia [9]. With hardly more than 25 taxa in the Southern Hemisphere, they are largely found in the Northern Hemisphere [53].

The genus is represented in Asia by about 350 species: 186 from China and about 180 from the former USSR [54], of which 45 are endemics. The Russian Federation contributes more than 80 species, and the European and Siberian parts of Russia, the Caucasus [54], and Kyrgyzstan contribute 54 species [55]. Uzbekistan contributes 47 species, 19 of which are endemic [56]. Tajikistan, contributes 48 species, only one of which is endemic [57]. Turkmenistan contributes 33 species, one of which is endemic [58]. Finally, Kazakhstan contributes 81 *Artemisia* species, 19 of which are endemic [59] (Table 1).

## 3. Traditional Use, Phytochemical Profiles and Pharmacological Properties of Artemisia Species from Central Asia

Modern phytochemical studies have shown that the *Artemisia* species from Kazakhstan contain structurally diverse secondary metabolites including sesquiterpenes, sesquiterpene lactone monomers and dimers of guaianolide, seco-guaianolide and eudesmanolide types, lignans, flavonoids, coumarins and volatile oils which show various health benefits [60,61]. A number of species of *Artemisia* in Kazakhstan have been explored by their chemical compositions and/or bioactivities, including *Artemisia altainsis*, *Artemisia austriaca* Jacq., *A. cina*, *Artemisia*
*leucodes* Schrenk, *Artemisia gmelinii* Weber, *Artemisia glabella* Kar. & Kir., *Artemisia succulenta* Ledeb., *Artemisia tschernieviana* Besser, *Artemisia frigida* Willd., *Artemisia filatovae* Kuorijanov, *Artemisia lerchiana* Weber, *Artemisia rupestris* L., *Artemisia halophila* Krasch. *Artemisia sieversiana* Ehrh., *Artemisia porrecta* Krasch. ex Poljakov, *Artemisia kasakorum* (Krasch.) Pavlov, *Artemisia radicans* Kupr., *Artemisia latifolia* Ledeb., *Artemisia pauciflora* Weber, *Artemisia pontica* L., *Artemisia hippolyti* Butkov, *Artemisia santolinifolia* Turcz. ex Bess., *Artemisia commutata* Besser and *Artemisia glauca* Pall. ex Willd (Table 2 and Table 3).

Until now, a detailed chemical study of the *Artemisia* flora of Kazakhstan as a promising source of biologically active substances has not been conducted, so their study is an urgent task. The most investigated biologically active compounds from the *Artemisia* species are sesquiterpene lactones, and searching for biologically interesting and structurally unique sesquiterpene lactones is a pressing topic in the field of natural products. Our research group, within the framework of the research projects of the Ministry of Education and Science of the Republic of Kazakhstan (AR05133199, AP08856717), has been analyzing and determining the main chemical composition of various *Artemisia* species such as *Artemisia absinthium* L., *Artemisia diffusa* Krasch. ex Poljak., *Artemisia rutifolia* Stephan ex Spreng., *A. nitrosa*, *Artemisia marschalliana* Spreng., *A. albicerata, Artemisia sublessingiana* Krasch. ex Poljakov, *A. pauciflora, Artemisia heptapotamica* Poljak., *Artemisia serotina* Bunge, *Artemisia terrae-albae* Krasch., *Artemisia scopiformis* Ledeb. and *Artemisia transiliensis* Poljakov from the Almaty and East Kazakhstan regions. The most interesting endemic and TKM-used Artemisia species were investigated comprehensively (Table 2 and Table 3).

*Artemisia glabella* Kar. & Kir. (smooth wormwood) is a perennial endemic plant growing only on the dry hills of Central Kazakhstan in the Karaganda region. From the rhizome of this species, several secondary metabolites, including the sesquiterpene lactones argolide and arglabin (Figure 2), were isolated [60]. Other phytochemical studies have identified the flavonoids, luteolin, bonansine, pectolinarigenin, cirsilineol and casticine (Table 3). The above-ground parts of *A. glabella*, i.e., the leaves, buds, flowers and stems, contain arglabin, a sesquiterpene lactone of the guaianolide type with two five-membered rings trans-annulated (Figure 2). Arglabin shows antitumor activity against different tumor cell lines [61]. Currently, arglabin is used in oncology clinics in Kazakhstan and other countries for the treatment of liver, ovarian, cervical, lung and breast cancers, either as monotherapy or in combination with other chemotherapeutic agents and radiation therapy. The method of obtaining arglabin has been patented in 12 countries including Japan, China, USA, Great Britain, Germany, Switzerland, France, Austria, Italy, the Netherlands and Sweden [62]. The arglabin preparation has passed randomized clinical trials in accordance with the international standards of clinical practice (GCP) and has been registered and developed as the first plant origin antitumor drug in Kazakhstan. Arglabin has been registered as an antitumor medicine in the Russian Federation, the Republic of Kazakhstan, Uzbekistan, Tajikistan, Kirgizstan and Georgia. Despite its vital therapeutic value, arglabin isolated from *A. glabella* accounts only for 0.27% of the aerial parts of the plant [63].

*Artemisia cina* O.C. Berg et C.F. Schmidt (in Kazakh language, “dermene”) is an aromatic plant which is listed in the Red Book of Kazakhstan (a document with complete information about all kinds of living beings and plants that are rare or are on the verge of extinction). It is endemic to South Kazakhstan territory, in the desert and lowland foothill areas, mainly on fertile, moist loamy and serozemic soils. It was used for many years in medicine to treat parasitological diseases. Its inflorescences have been used since ancient times by Kazakhs for the treatment of helminthiasis, asthma and bronchitis [11]. *A. cina* is a source of the sesquiterpene lactone, α-santonin and artemisinin (Figure 2). Santonin is a parasitic agent; it has a detrimental effect on roundworms and a weaker effect on other worms [63]. The essential oil from *A. cina* lowers blood pressure, narrows blood vessels, lowers the tone of smooth muscles, is effective in the treatment of rheumatism and neuralgia, has an anti-inflammatory effect, weakens allergic reactions, enhances regenerative processes and helps in the treatment of eczema and X-ray burns [63].

*Artemisia**leucodes* Schrenk (whitish wormwood) is a weed that occurs in the zone of the sandy deserts of Kazakhstan. Aterolid is a preparation of the aerial parts of *A. leucodes* and is currently in clinical trials in Kazakhstan for the treatment of arteriosclerosis [64,65]. Chemical investigations of *A. leucodes* have shown the presence of sesquiterpene lactones including austricin, grossmizin, anhydroaustricin and matricarin [66] (Table 3). The essential oil of *A. leucodes* has been reported to present antiseptic, local analgesic and anti-inflammatory effects [67], where the major components are camphor, camphene and 1.8-cineole (Table 2).

*Artemisia sieversiana* Ehrh. (Sage sivers/wormwood sivers) is found in the Central part of Kazakhstan, the Far East and Western and Eastern Siberia. *A. sieversiana* is endowed with very valuable therapeutic properties; it is recommended to use the roots and aerial parts of this plant for various medicinal purposes. The associated therapeutic properties are mostly attributed to the presence of ascorbic acid, organic acids, coumarins, flavonoids, carotenes and γ-lactone. Sesquiterpene alcohols are the main components in its essential oil (Table 2) [68]. Infusions and decoctions prepared on the basis of the inflorescences, roots and aerial parts of this plant are used in both folk and Tibetan medicine for bronchitis and cough. As a diaphoretic, an infusion based on the herb wormwood sivers is used for fever and colds. This plant has the ability to improve appetite and the activity of the gastrointestinal tract, can be used as an antihelminthic agent and is successfully used for constipation. The plant’s essential oils also possess antifungal, antihelminthic, antimicrobial and anti-inflammatory activity in vivo due to the presence of azulenes [69,70].

*Artemisia lercheana* Weber ex Stechm (Lerch sage) perennial herb grows on the sandy soils and steppe regions of the Aktobe, Atyrau, West Kazakhstan, Kostanay, Karaganda and Almaty regions of Kazakhstan [59,69]. The essential oil composition of *A. lercheana* was also reported (Table 2), where ~109 compounds were identified in the essential oil. The principal components were β-thujone, 45.6%; α-thujone, 24.2; camphor, 7.5; and 1,8-cineol, 4.6 [69].

*Artemisia albida* Willd. is a semi-white wormwood that grows in the Altai, Tarbagatai, Zaysan and Akmola regions of East Kazakhstan [71]. From 2005 to 2008, *A. albida* Willd, collected from the Ivanovsky ridge (Altai) in East Kazakhstan, was first studied by S.M. Adekenov and E.M. Suleimenov. Eupatylin and its 5’-methyl ester, as well as the sesquiterpene lactones of Austrian, Matricaria, Canin and Argolide, have been isolated from *A. albida* (Table 3). In addition, for the first time, Anhydroaustricin, two eupatilin flavonoids and its 7-*O*-methyl ester were isolated from the aerial part of *A. albida* Willd [72,73].

*Artemisia heptapotamica* Poljakov (syn. of *Seriphidium heptapotamicum* (Poljakov) Ling & Y.R. Ling) is an endemic medicinal plant of Kazakhstan, mainly distributed in the north part of the Tian Shan Mountains that are found in Dzungarian Alatau, Ili, Kungei Alatau and Ketmen. A new dimeric and two new monomeric sesquiterpene lactones, together with three dimers and seven monomers (guaianolides and *seco*-guaianolide skeleton), were isolated and identified. Most of the isolated monomeric sesquiterpenes showed strong inhibitory actions against the lipopolysaccharide (LPS)-induced NF-κB activation on a THP1-dual cell model, with IC_50_ values ranging from 2 to 25 μM (Table 3) [74].

*Artemisia rupestris* in Kazakh culture is known as “Kyeli-Ermen” [6]. It is well known in TKM to reduce fever in infectious and inflammatory diseases, improve the function of the gallbladder, reduce inflammation, relieve inflammation of the stomach and relieve nausea and upset stomach. It is used mainly in diseases such as: dry stomach, upset stomach, inflammation of various joints, fever, antitumor and anti-anaphylaxis [75,76]. It is a rich source of rupestonic acid, which has antiviral activity. The studies showed that rupestonic acid exhibits significant activity against influenza type B viruses [77]. *A. rupestris* effectively suppresses inflammatory responses and antibacterial, anti-inflammatory and hemostatic activities [78]. In ancient Kazakh life, *A. rupestris* was prepared to be drunk as tea to treat certain diseases such as cancer, stomach pain, indigestion, jaundice, flu, urticaria and various types of hepatitis [79]. The crude extract obtained from *A. rupestris* tea by evaporation is usually used to treat skin diseases such as neurodermatitis, poisonous insect bites and all types of skin lesions [6,80].

*Artemisia frigida* Willd is also known in TKM as “sacred kermek”. The medicinal part of this plant is the aboveground part. It is mainly used as a remedy for stomach discomfort, inflammation of the internal organs, skin diseases and inflammation and to improve the functioning of the gallbladder [6,80,81].

*Artemisia annua* is commonly found in several areas of Kazakhstan and Central Asia. The decoction of its aerial part has been used in TKM for its antipyretic properties and is also used to treat indigestion, constipation and skin diseases [6,7,8]. It is known worldwide due to the isolation of the well-known antimalarial agent artemisinin. This sesquiterpene lactone belongs to the cadinanolide type with a rearrangement to the notorious 1, 2, 4-trioxane ring and the typical endoperoxide system (Figure 2). Its discovery in 1972 and the further development of artemisinin as an antimalarial agent led Dr. Tu Youyou being a co-recipient of the 2015 Nobel Prize in Medicine [82].

*Artemisia transiliensis* Poljakov is an endemic species spatially limited by the loess foothills of the Zaili Alatau in the Alma-Ata region, predominantly living on loamy dark chestnut soils. This type of wormwood has long been used as an anti-germ agent. The essential oils of *A. transiliensis* have a strong bactericidal, anti-inflammatory and analgesic effect. The chemical composition of *A. transiliensis* is rich in sesquiterpenes, steroid hormones, mineral elements and essential oils. Our research group investigated the qualitative and quantitative phytochemical contents, including the determination of amino and fatty acids in the aerial parts of the plant *A. transiliensis* [83,84]. Raw materials were collected at the flowering phase from the Almaty region in 2018.

*Artemisia serotina* Bunge. (winter wormwood) is a perennial shrub 40–80 cm high, endemic to Central Asia. It is a plant growing in Central Asia, on the territory of Kazakhstan and Uzbekistan. This plant grows on various types of soils, sometimes in slightly saline places in the plains and foothills and less often in the lower belt of the mountains. It grows in the desert zone on saline clayey and sandy loamy soils, river terraces, dry sai, gravel-clay slopes of foothills and as a weed on pastures, fallow lands, abandoned arable lands and near roads [85]. A general analysis of the essential oils isolated by *A. serotina* was carried out by the S. Yu. Yunusov Institute of the Chemistry of Plant Substances, Academy of Sciences of the Republic of Uzbekistan. As a result of the studies carried out by the method of the chromato-mass spectral analysis of the benzene extract of the aerial part of *A. serotina* growing in Uzbekistan, 22 compounds were identified for the first time. The main components are 1,8-cineol—10.08%, filifolid A—8.62%, chrysanthenon—13.00% and (Z)-jasmone—1.95% [86]. In comparison with the data obtained, *A. serotina*, growing in Kazakhstan, showed that the main components of essential oils are thujone—53.9%, carvone—5.7%, camphor—2%, 1.8-cineol—6.6%, isobutyric acid—0.03% and phenols—0.05%. Additionally, seven glucosides and six aglycone flavonoids were isolated from the MeOH extract. Flavonoid aglycones were identified on the basis of physicochemical properties, comparative chromatography and UV and IR spectra [87]. According to a study conducted in Uzbekistan in 2019, the plant contains cineole (78%), sesquiterpendilactones. 5-monoterpenoids (thujone, caravan, camphor, 1,8-cineole, non-alcoholic) were isolated from the essential oil of *A. serotina* [88].

*Artemisia scopaeformis* Ledeb. is an unexplored endemic species. The chemical composition of this plant was studied at the Research Center of Medicinal Plants. The condensed *A. scopaeformis* crude ethanol extract was used for preliminary qualitative and quantitative screenings and for the identification of bioactive chemical constituents such as alkaloids, saponins, flavonoids, polysaccharides, coumarins, organic acids including moisture, ash and extractives. Moreover, from ash of the plant, an elemental analysis by atomic absorption spectrophotometry was conducted, and it showed the presence of both macro and micronutrients. The plant extract was further investigated for fatty and amino acid compositions. The most dominant fatty acid in *A. scopaeformis* was unsaturated fatty acid – linoleic acid, which, along with glutamate, is one of the most abundant in amino acids [89]. Crude extract was obtained from the aerial part of *A. scopaeformis* through selective sequential extraction with solvents of increasing polarity, namely, n-hexane and chloroform. GC–MS analysis of the n-hexane and chloroform extracts revealed the presence of various bioactive compounds which can be responsible for antioxidant, antifungal, antimicrobial, anti-inflammatory and anticancer potentials [90].

*Artemisia aralensis* Krasch. is an endemic high vascular plant which is an inhabitant of the northern shore of the Aral Sea in the Syr-Darya river bottom. Prof. Adekenov at al. identified the chemical composition of the essential oil and quantitative contents of the terpenoids for the first time (Table 2 and Table 3) [91].

It is significant to highlight, in a brief comment, the importance of the *Artemisia* species that have been shown through centuries as playing a critical role in dealing with serious human diseases, and currently, the attention has been turned to the use of this important genus on viruses, including the coronavirus disease (COVID-19). Zhou et al. reported the in vitro efficacy of *Artemisia annua* extracts as well as artemether, artesunate and artemisinin against SARS-CoV-2 via the immunostaining of SARS-CoV-2 spike glycoprotein [92]. Remarkable therapeutic inhibitions were recorded for the SARS-CoV-2 infections of VeroE6 cells, human hepatoma Huh7.5 cells and human lung cancer A549-hACE2 cells, where artesunate was shown to be most active (EC_50_ 7–12 μg/mL) [93].

*Artemisia sublessingiana* Krasch. ex Poljakov is native to Altay, Kazakhstan, Kirgizstan, Mongolia and Xinjiang. In the 1970–80s, the antimicrobial and antitumor activity of the phenolic composition was investigated, including the structure of the new sesquiterpene lactone arsubin from *A. sublessingiana* [94,95]. Later, R. Jalmakhanbetova et al. investigated the ethanol extract of *A. sublessingiana* aerial parts, which led to the isolation of six flavonoids and a sesquiterpenoid. The isolated compounds were (1) eupatilin, (2) 3,4′-dimethoxyluteolin, (3) 5,7,3′-trihydroxy-6,4,5-trimethoxyflavone, (4) hispidulin, (5) apigenin, (6) velutin and (7) sesquiterpene lactone 8α,14-dihydroxy-11,13-dihydromelampolide. The isolated compounds were in silico examined against the COVID-19 main protease (Mpro) enzyme. Compounds 1–6 exhibited promising binding modes showing free energies ranging from −6.39 to −6.81 (kcal/mol). The best binding energy was for compound (2) [96].

*Artemisia commutata* Besser is a perennial herb found in northern and eastern Kazakhstan. The plant is also found in Europe, Mongolia, Altai, Western and Eastern Siberia and the European part of Russia [81]. A new flavonoid, Jusanin, has been isolated from the aerial parts of *A. commutata* by Yerlan M. Suleimen et.al. Jusanin showed a high structural similarity degree with X77, the co-crystallized legend of the COVID-19 main protease (PDB ID: 6W 63) Mpro. This result was indicated by molecular similarity, fingerprint and DFT studies. The molecular docking of 1 against Emperor confirmed the correct ending of 1 insidePro, exhibiting a binding energy of −19.54 Kcal/mol. The ADME and toxicity properties of 1 indicated its likeness to be a drug as well as its general safety. The MD simulation studies at 100 ns confirmed the correct binding of the Mpro–Justinian complex. These interesting results open the door to finding a treatment against COVID-19 after in vitro and in vivo studies. In addition, three other metabolites have been isolated and identified: capillartemisin A, methyl-3-[S-hydroxyprenyl]-cumarate and *β*-sitosterol [97].

*Artemisia glauca* grows in the territory of Northern and Eastern Kazakhstan and is also distributed in Mongolia, Siberia, Europe and North America [98]. A new dicoumarin, jusan coumarin, was isolated from *A. glauca* aerial parts by Yerlan M. Suleimen et.al. Jusan coumarin demonstrated a high degree of similarity with X77, the co-crystallized ligand of Mpro. The similarity was confirmed by four ligand-based computational, molecular similarity, fingerprint, DFT and pharmacophore studies. The molecular docking studies of 1 against Mpro verified the perfect binding of 1 inside the active site of Mpro, exhibiting a binding energy of −18.45 kcal/mol. The ADME and toxicity profiles of 1 showed its overall safety and its possibility to be used as a drug. The MD simulation studies authenticated the binding of 1 inside the Mpro. These findings give hope to find a cure for COVID-19 upon further in vitro and in vivo studies. Additionally, the known coumarin derivative 7-isopentenyloxycoumarin has been isolated, along with sitosterol. [98].

The investigation results of the phytochemical profiles and pharmacological activities of *Artemisia* species from Kazakhstan and Central Asia are shown in Table 2 and Table 3. The *Artemisia* species revealed the presence of mono-, di-, sesqui- and triterpenoids, sesquiterpene lactones, phenolic compounds, flavonoids and lignans (Table 3). It has been found that these constituents possess pharmacological activities such as anthelmintic, antimicrobial, anti-inflammatory, antitumor, antioxidant, cytostatic, antifungal, antimalarial, antileishmaniasis, antinociceptive, immunomodulatory and antipyretic activities, as well as strong inhibitory activity against FPTase [24]. The most investigated biologically active compounds from the *Artemisia* species are the sesquiterpene lactones, and the search for biologically interesting and structurally unique sesquiterpene lactones is still an active topic in the field of natural products.
molecules-27-05128-t002_Table 2Table 2Major components of essential oils from *Artemisia* species of Central Asia [6,67,69,70,74,75,80,81,86,89,90,99,100,101,102,103].Chemical Constitutes/*Artemisia* speciesContent, %*A. terraealba**A. frigida**A. glabella**A. rupestris**A. filatovae**A. lercheana**A. sieversiana**A.hyppolyti**A. armeniaca**A. proceriformis**A. dracunculus**A. marschalliana**A. gmelinii**A.kasakorum**A. leucodes**A. serotina**A.aralensis***α-pinene**- *1.6-2.00.90.11.7-3.4-4.24.10.10.15.3--**Camphene**9.85-0.40.30.80.9---0.4--0.39.31--**Sabinene**-0.650.13.00.20.6-2.11.320.20.80.20.1---**Myrcene**---9.518-14.2-6.2-6.20.6--0.30--**α-terpenene**-1.1-0.30.3-0.1---3.10.21.00.3---**p-cymene**----1.5-3.4-3.50.61.52.13.73.4---**Limonene**-0.3----0.40.727.7-3.31.0-----**1, 8-cineole**23.924.7120.34.34.69.323.61.13.10.313.528.5-6.206.613.19**Artemisia ketone**-----------4.4-----**Linalool**-0.480.20.30.14.29.87---0.70.10.2---**α-thujone**-5.2-0.1-24.2---66.3-5.78.62.7-53.9-**β-thujone**6.01.3-2.5---45.60.1--23.4-1.41.0
---**Camphor**47.322.60.23.6-7.5-1.47---9.811.33.739.02-**Borneol**-3.95.20.3---6.15---3.39.3

--**Terpene-4-ol**-2.86.51.90.40.7-1.421.42.0-2.63.32.50.53-4.38**Cumin aldehyde**--9.4-0.50.3------0.32.4---**β-elemene**---5.411-2.0-6.9-1.10.4-----**Spathulenol**--2.43.82.00.22.1
30.4-3.03.50.51.7

3.27**Trans-3 (1-butenyl)-isocoumarin**----------10.3------**Capric acid**---5.1--
----------**γ-costol**---3.9--0.4----------**Valencene**---3.71.42------------**Cumin alcohol**--3.7--0.2-----------**Nerolidol**----14------------**Pinan-2-ol**----8.3------------**Achillene**----5------------**Neryl isovalerate**------3.4----------**Caryophyllene**------3.0----------**Carvone**------
--------5.7-* (-) means not identified.
molecules-27-05128-t003_Table 3Table 3Secondary metabolites isolated from *Artemisia* L. species of Central Asia.№PlantsStructureChemical FormulaCompoundActivity*1**Artemisia cina* O.C. Berg et C.F. Schmidt
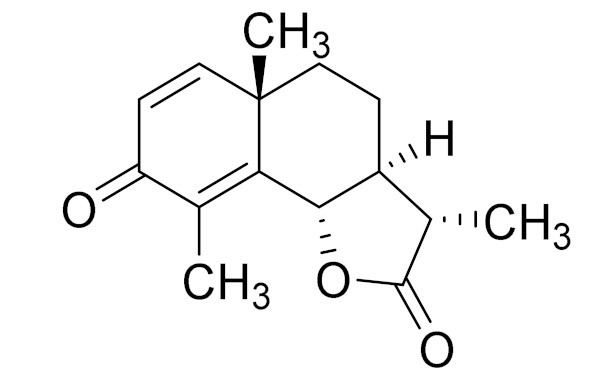
C_15_H_18_O_3_α-Santonin [63]Anthelmintic [63] andantipyretic activity [104]*2**Artemisia tschernieviana*Besser
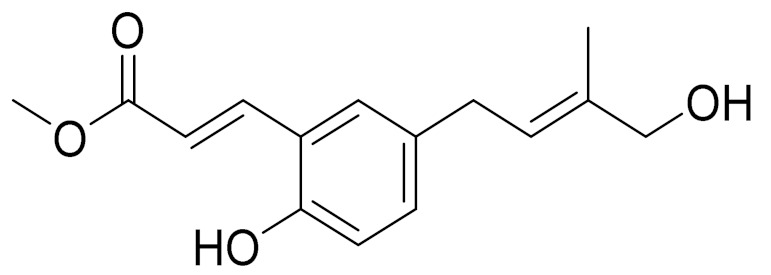
C_15_H_18_O_4_Methyl ester of 3-(5_hydroxyprenyl)-*p*-coumaric acid [103]-*3**Artemisia sieversiana* Ehrh.
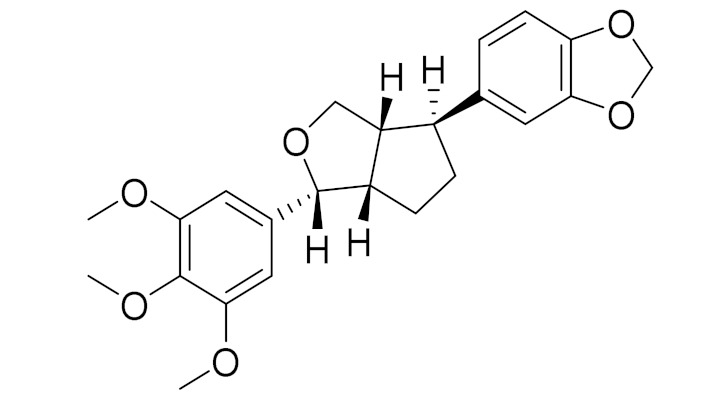
C_22_H_24_O_7_4-Epiashantin [105]Reasonable antimicrobial activity toward the aforementioned microorganism strains [105]*4**Artemisia heptapotamica*Poljak.
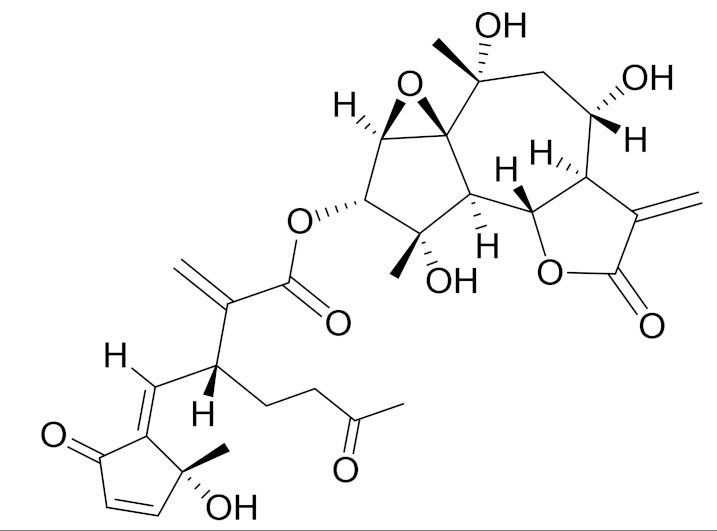
C_31_H_37_O_13_Artemisiane E [74]-
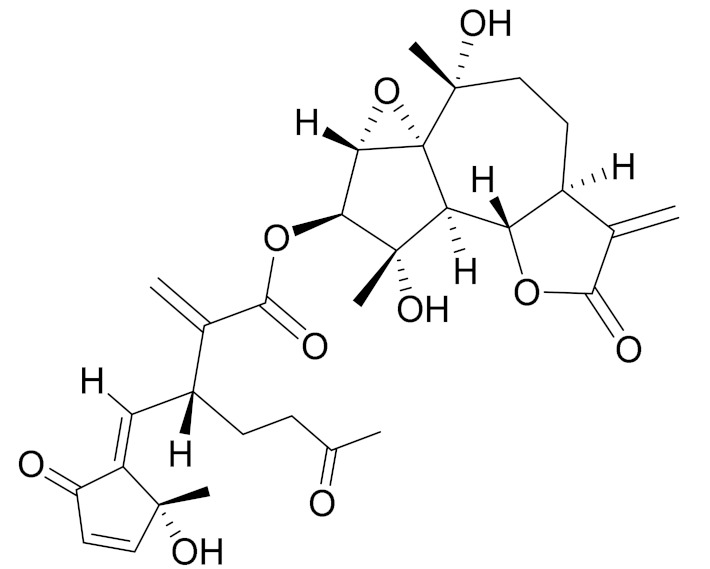
C_30_H_36_O_10_Artemisiane A [74]-
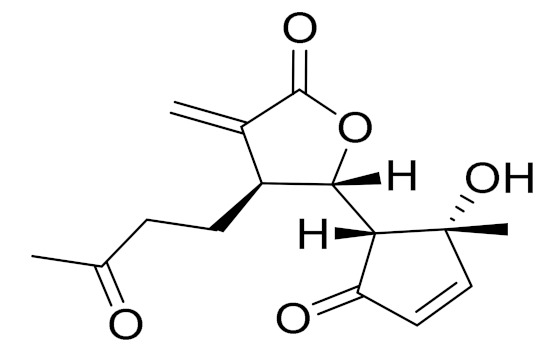
C_15_H_18_O_5_*seco*-Tanapartholide A [74]Anti-inflammatory effects [74]
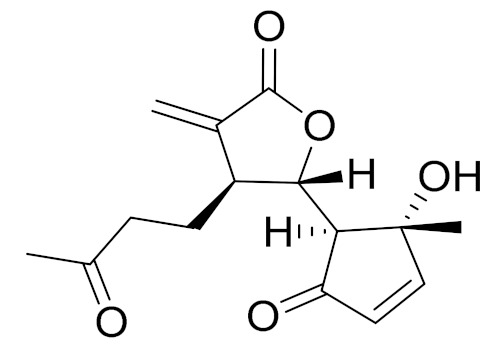
C_15_H_18_O_5_5-*epi*-Secotanapartholide A [74]Anti-inflammatory effects [74]
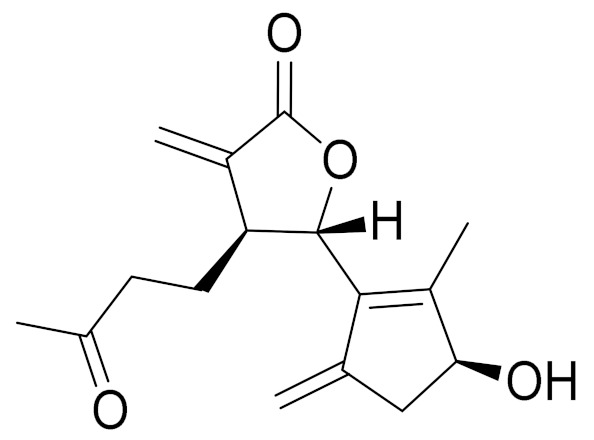
C_16_H_20_O4*iso*-seco-TanapartholideAnti-inflammatory effects [74]
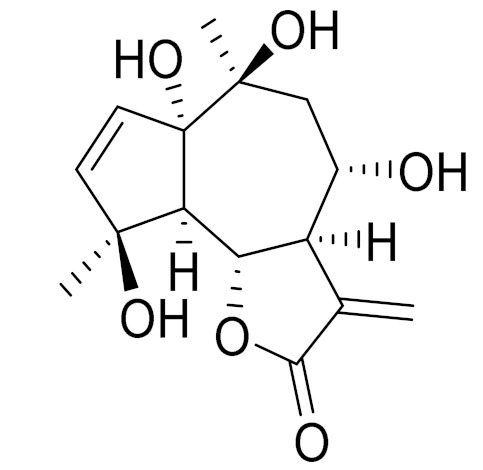
C_16_H_21_O_8_Artemdubolide I.-
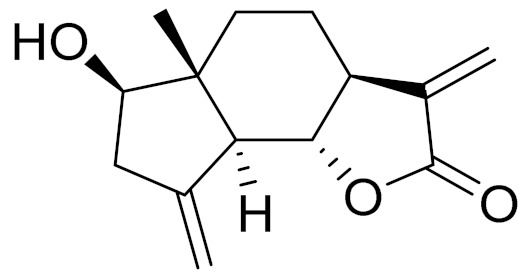
C_14_H_17_O_3_Ajaniaolide BAnti-inflammatory effects [74]
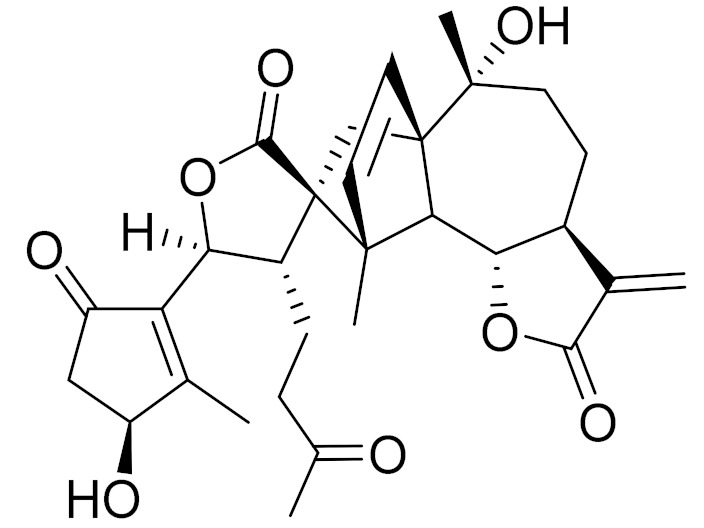
C_30_H_36_O_8_3*β*-chloro-4*α*, 10*α*-dihydroxy-1*α*,2*α*-epoxy-5*α*, 7*αH*-guaia-11(13)-en-12,6*α*-olideAnti-inflammatory effects [74]
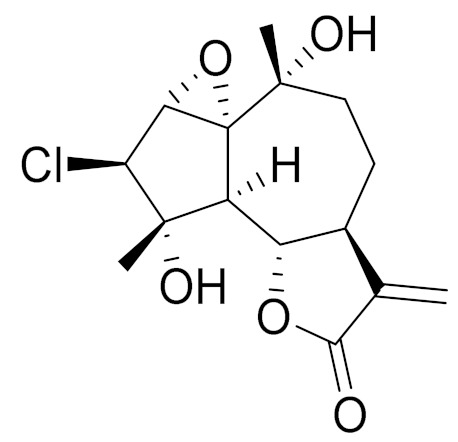
C_15_H_19_ClO_5_3*α*-chloro-4*β*,10*α*-dihydroxy-1*β*,2*β*-epoxy-5*α*,7*α* Hguai-11(13)-en-12,6*α*-olideAnti-inflammatory effects [74]
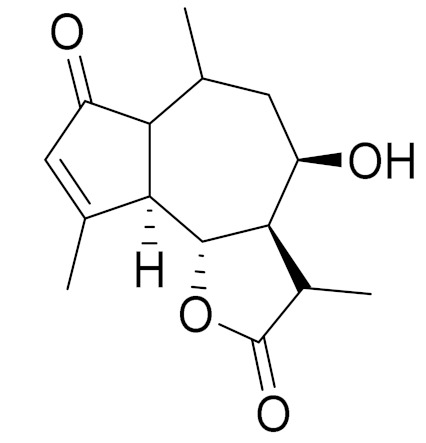
C_15_H_20_O_4_Rupicolin BAntimicrobial activity [106]
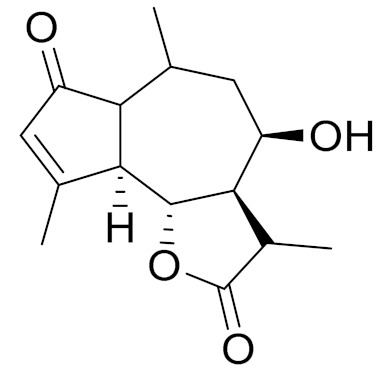
C_15_H_20_O_4_HydroxyachillinAnti-inflammatory activity incarrageenan-induced paw edema [107]
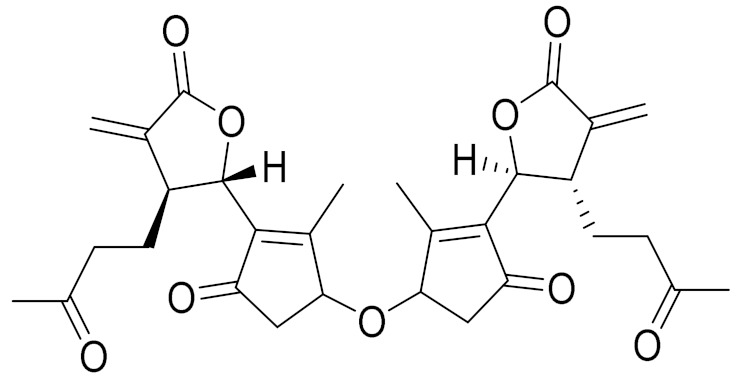
C_30_H_34_O_9_Millifolide ATested on the following tumor cell lines: MCF7, HL-60 and PC3; however, it did not exhibit any cytotoxicities [108]
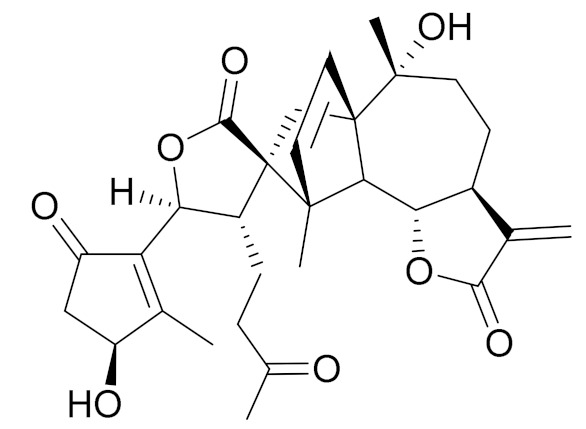
C_30_C_36_O_8_Achillinin CAntitumor agent [109]*5**Artemisia glabella* Kar. & Kir.
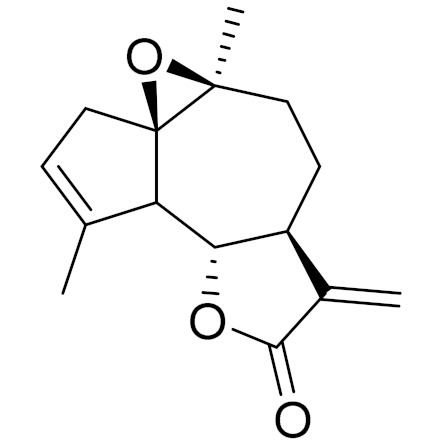
C_15_H_18_O_3_Arglabin [110,111]Antitumor activity [112]
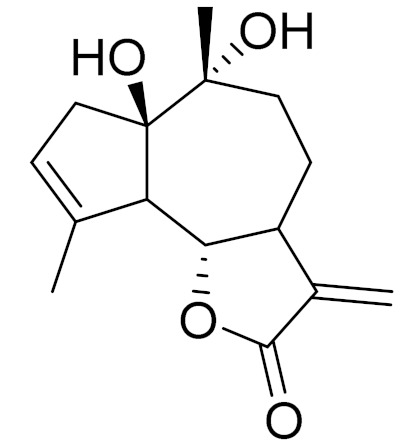
C_15_H_20_O_4_1β,10α-Dihydro-xyarglabin [113]-
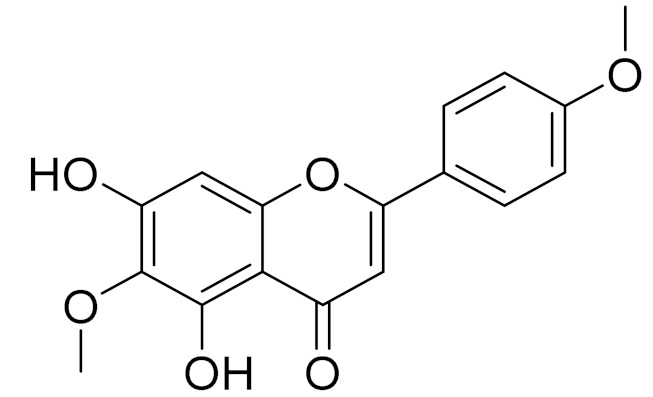
C_17_H_14_O_6_Pectolinarigenin [113]Anti-inflammatory effects [113]
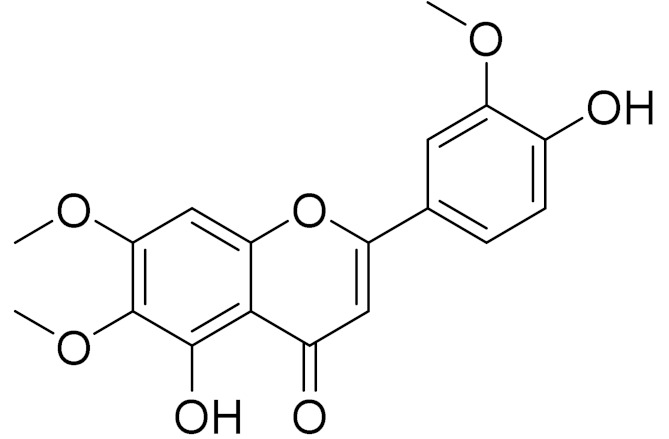
C_18_H_16_O_7_Cirsilineol [114]Antioxidant, cytostatic,antimicrobial, antifungal, antimalarial and antileishmaniasis activities [114]
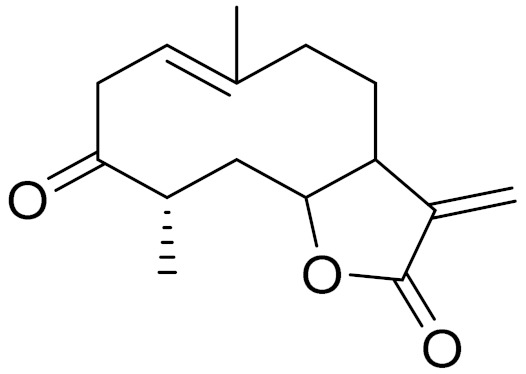
C_15_H_20_O_3_Argolide [113]Studied for analgesic activity; however it did not show any activities [115]
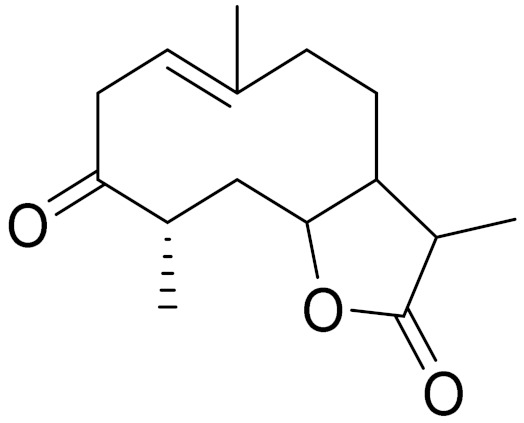
C_15_H_22_O_3_DihydroargolideModulate TCR activation, which is responsible in inflammatory and immune responses [116]*6**Artemisia halophila*Krasch.
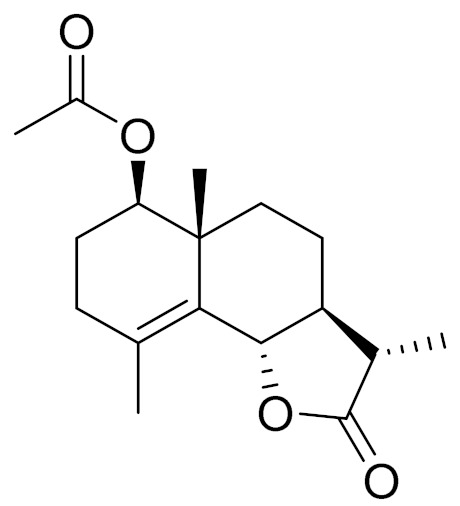
C_17_H_24_O_4_Arhalin [117]-
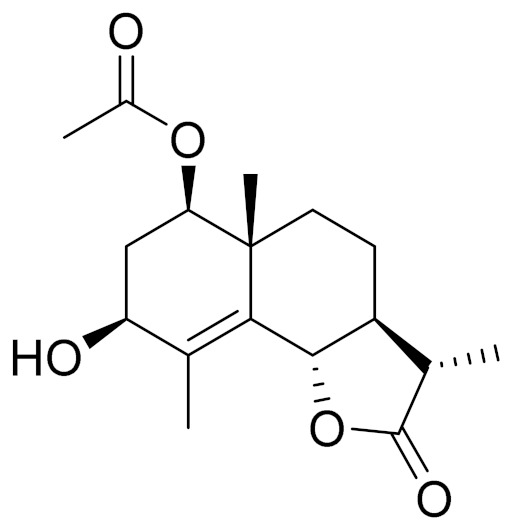
C_17_H_24_O_5_3-Hydroxyarhalin [117]Modulate TCR activation [118]
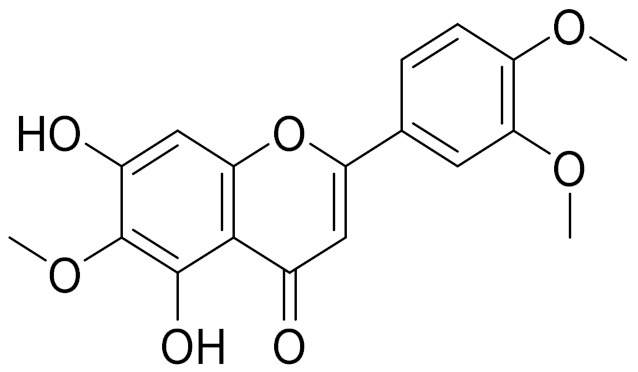
C_18_H_16_O_7_EupatilinAnti-inflammatory activity [119]*7**Artemisia semiarida* (Krasch. & Lavrenko) Filatova
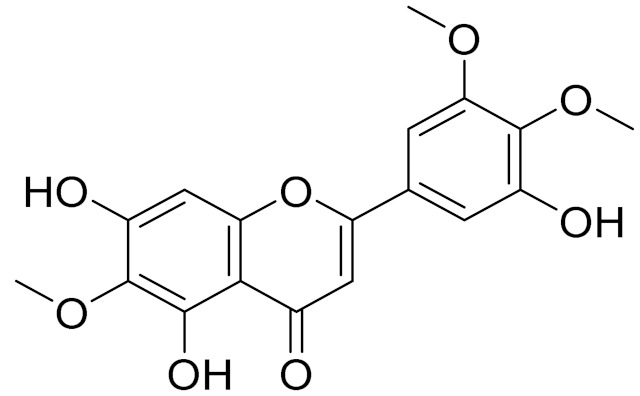
C_18_H_16_O_8_5,7,3-trihydroxy-6,4,5-trimethoxyflavone [120]Strong inhibitory activity against an FPTase [121]
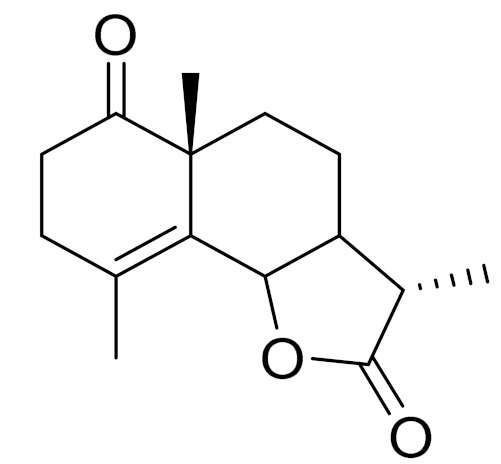
C_15_H_20_O_3_TaurinAntioxidant [122]
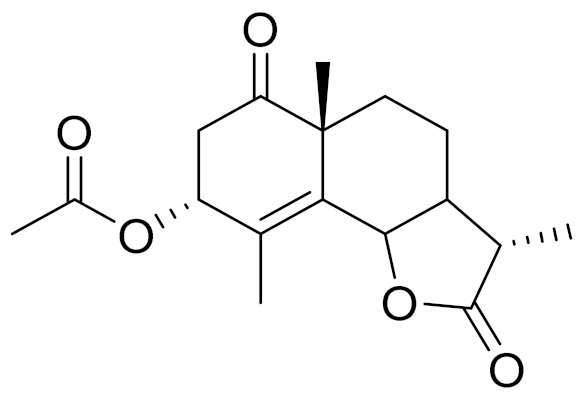
C_17_H_22_O_5_Acetoxytaurin-
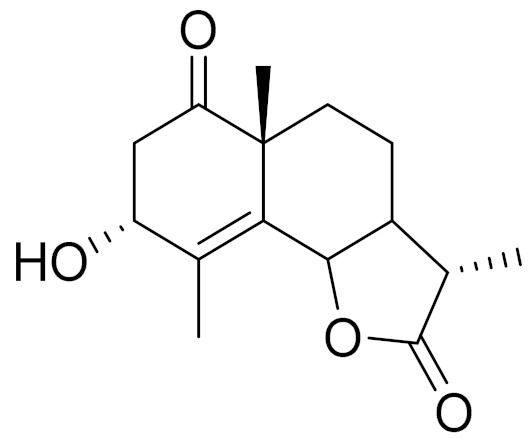
C_15_H_20_O_4_HydroxytaurinAntiprotozol effect against*Leishmania dolzovani* [122]
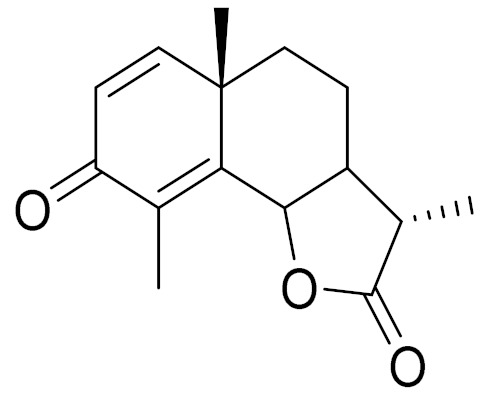
C_15_H_18_O_3_a-Santonin [63]Anthelmintic [63] andantipyretic activity [104]*8**Artemisia succulenta* Ledeb.
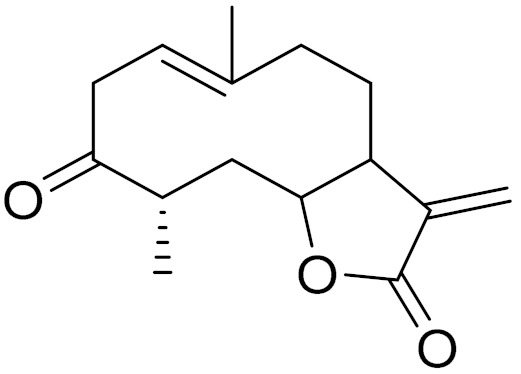
C_15_H_20_O_3_Argolide [123]Studied for analgesic activity; however, it did not show any activities [112]*9**Artemisia radicans* Kupr.
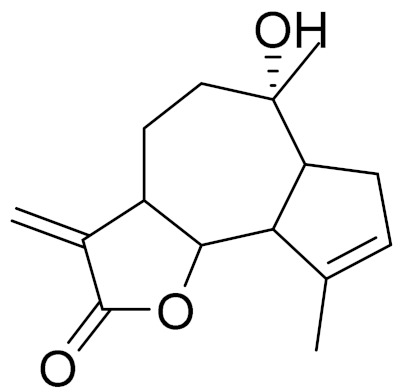
C_15_H_20_O_3_8-Deoxycumambrin [123]Aromatase inhibition [124]
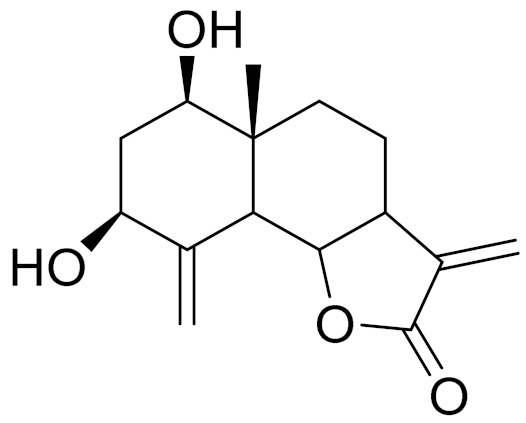
C_15_H_20_O_4_Ridentin B [123]Studied for action on human adherent cell lines but did not show any activities [125].*10**Artemisia filatovae* Kuorijanov
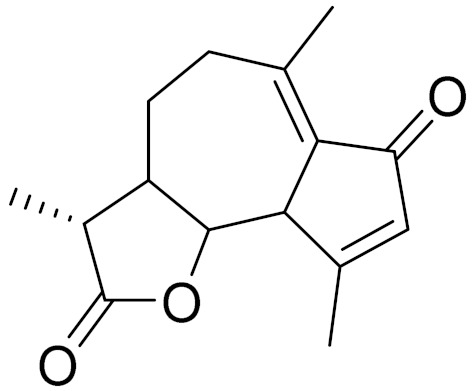
C_15_H_18_O_3_AchillinChemosensitizer agent [126]
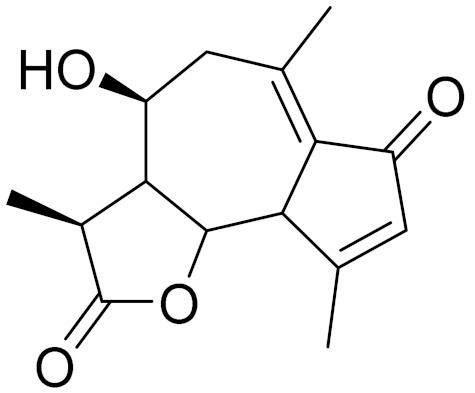
C_15_H_18_O_4_AustricinAngioprotector and antilipidemic activity [126]
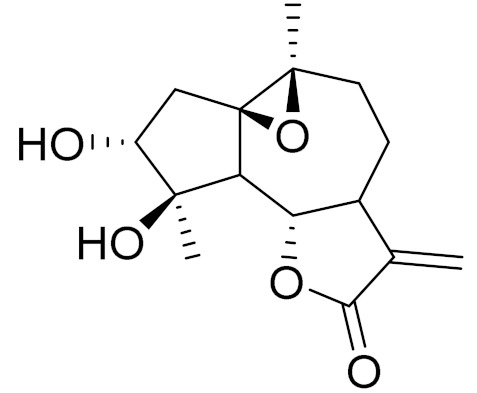
C_15_H_20_O_5_Artefin [127]Shows neurite outgrowth [128]*11**Artemisia porrecta* Krasch. ex Poljakov
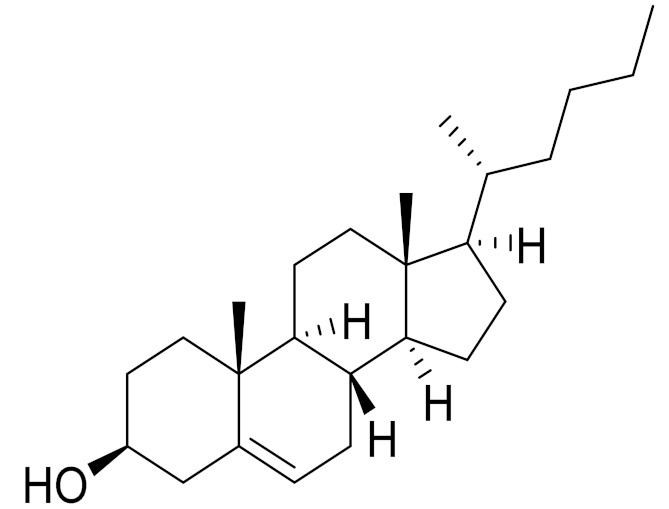
C_25_H_42_ONymphayol [129]Antinociceptive, immunomodulatory and antipyretic activity [130]
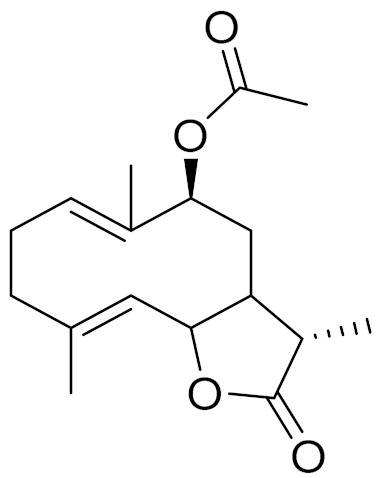
C_17_H_24_O_4_Gerbolide A-*12**Artemisia albida* Willd. ex Ledeb.
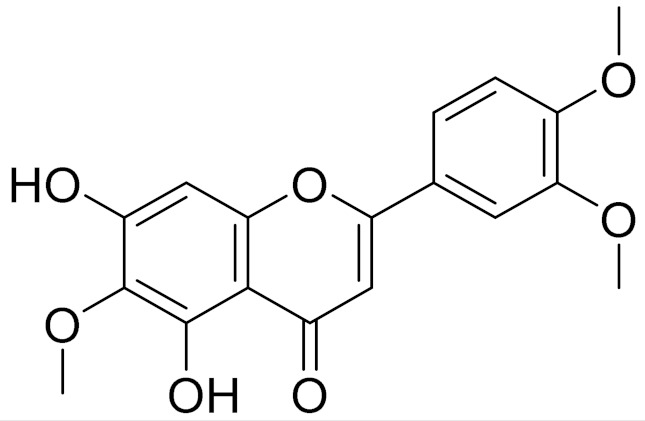
C_18_H_16_O_7_Eupatilin [71]Anti-inflammatory activity [119]
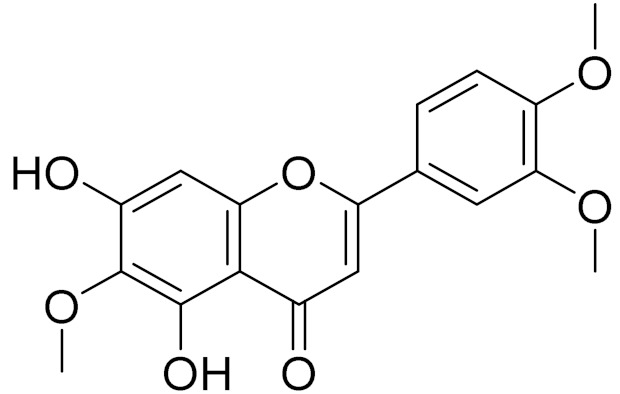
C_18_H_16_O_7_7-O-Methyl ester of eupatilin-
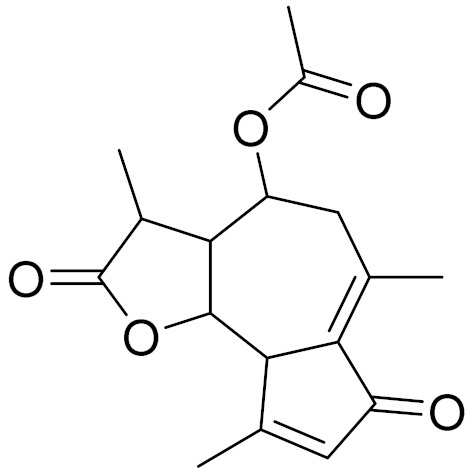
C_17_H_20_O_5_Matricarin-
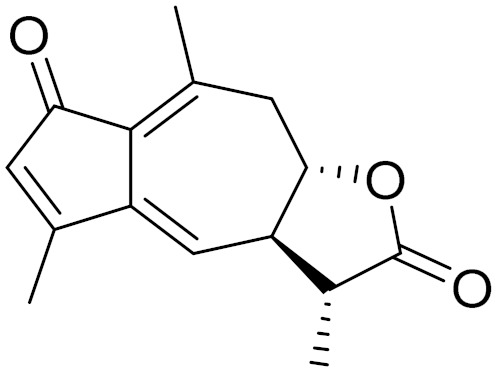
C_15_H_16_O_3_Anhydroaustricin [72]Low activity against malaria [126]
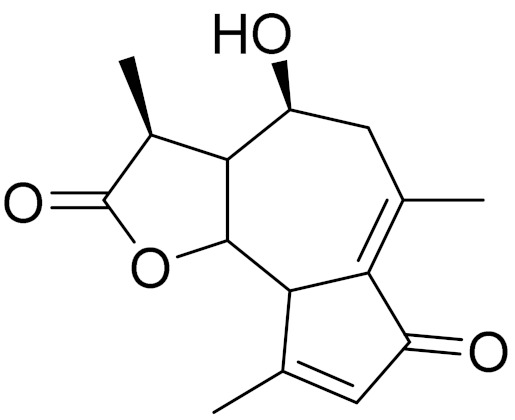
C_15_H_18_O_4_AustricinAngioprotector and antilipidemic activity [126]*13**Artemisia tournefortiana* Rchb.
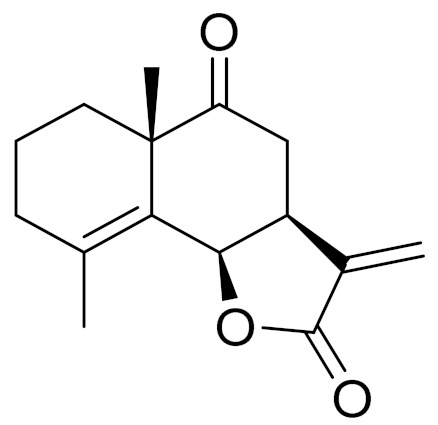
C_15_H_18_O_3_Taurneforin [131]-*14**Artemisia**pontica* L.
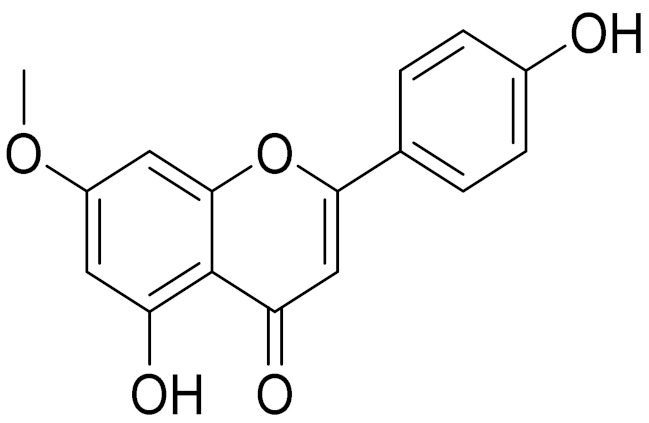
C_16_H_12_O_5_Genkwanin [132]Anti-inflammatory activity [133]
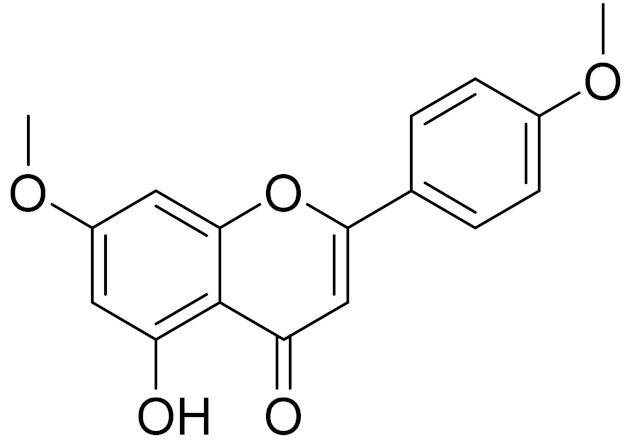
C_17_H_14_O_5_Apigenin 7,4’-dimethyl ether [134]-
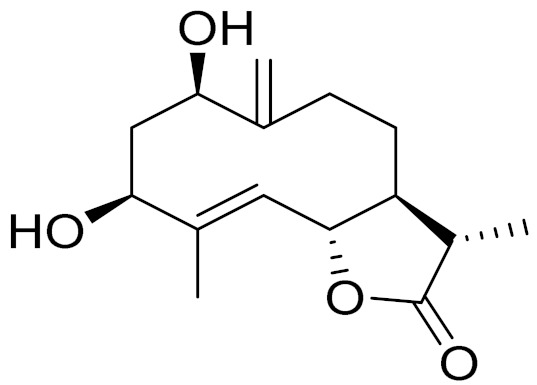
C_15_H_22_O_4_Dihydroridentin [134]-*15**Artemisia leucodes* Schrenk
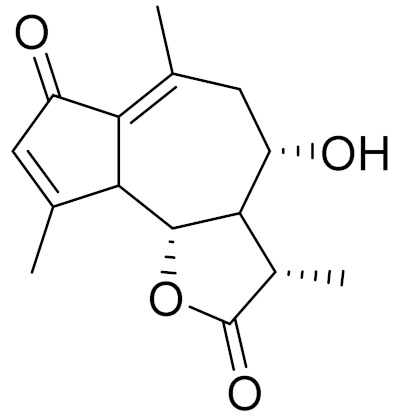
C_15_H_18_O_4_5-β(H)-Austricin [135]-
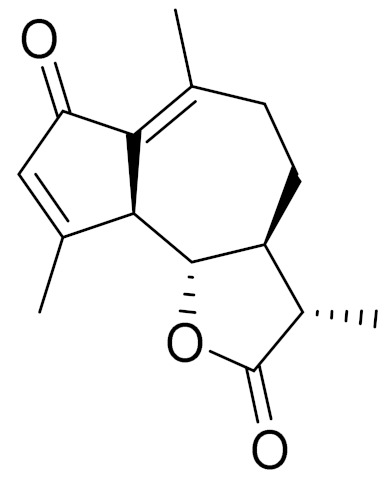
C_15_H_18_O_3_Leucomisin [136]Antibacterial and phagocytosis-stimulating activity [126]
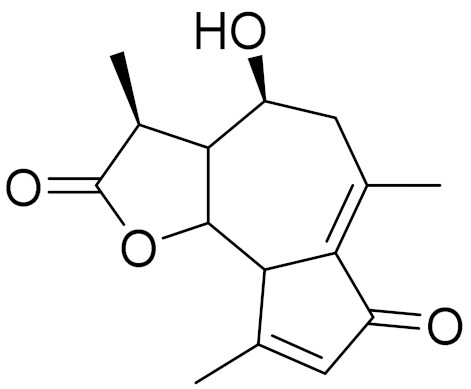
C_15_H_18_O_4_AustricinAngioprotector and antilipidemic activity [126]
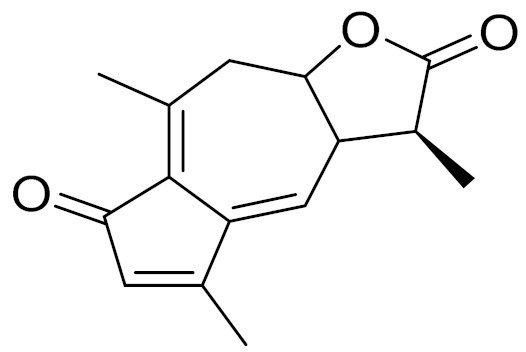
C_15_H_16_O_3_GrossmizinHypolipidemic activity [137]
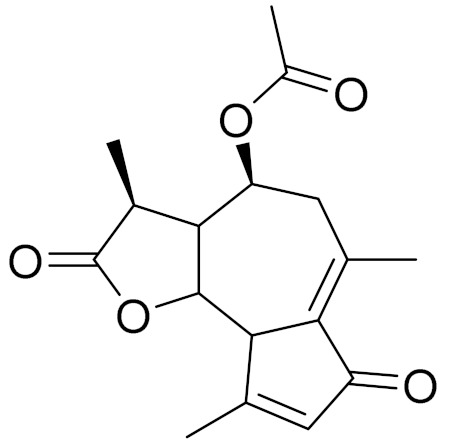
C_17_H_20_O_5_Matricarin-*17**Artemisia gracilescens* Krasch. & Iljin
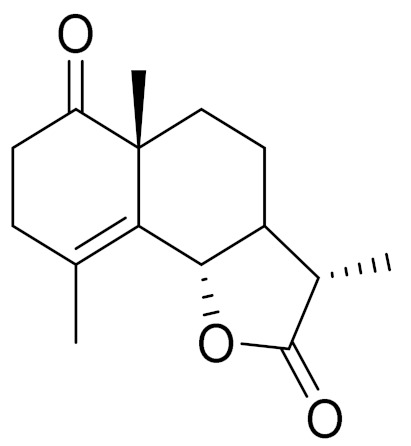
C_15_H_20_O_3_Gracilin [138]Immunosuppressive activity [139]
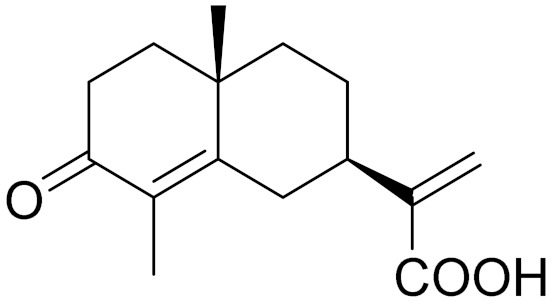
C_15_H_20_O_3_3-oxocostusic acid [140]Antibacterial activity [140]
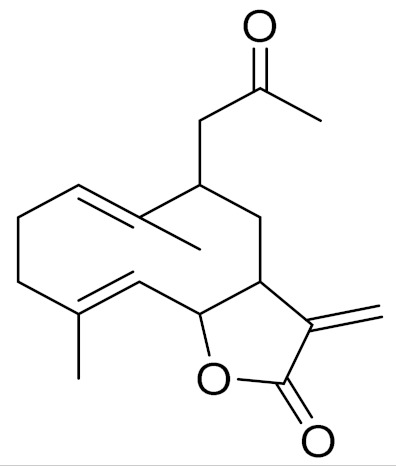
C_18_H_24_O_3_Argracin [141]TCR activity [141]*18**Artemisia subchrysolepis* Filat.
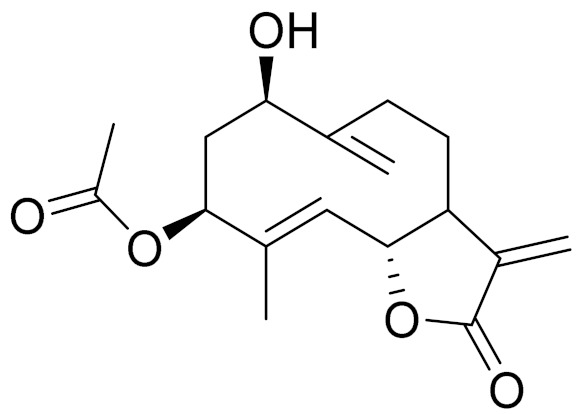
C_17_H_22_O_5_Subchrysin [142]-*19**Artemisia altainsis*
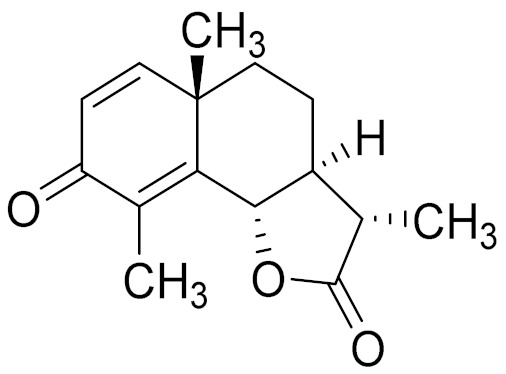
C_15_H_18_O_3_α-Santonin [63]Anthelmintic [63] andantipyretic activity [104]


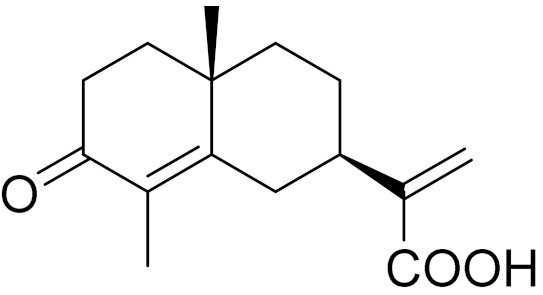
C_15_H_20_O_3_3-Oxocostus acid [140]Antibacterial activity [140]
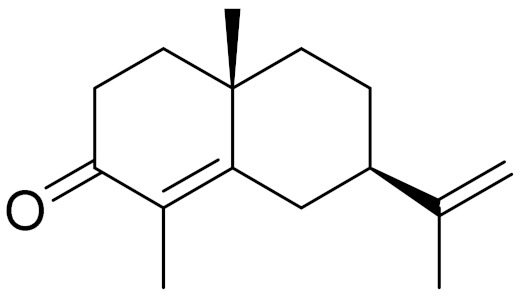
C_15_H_22_Oα-CyperoneAntivirulence, antigenotoxic and antibacterial activities [141]*20**Artemesia austriaca* Jacq.
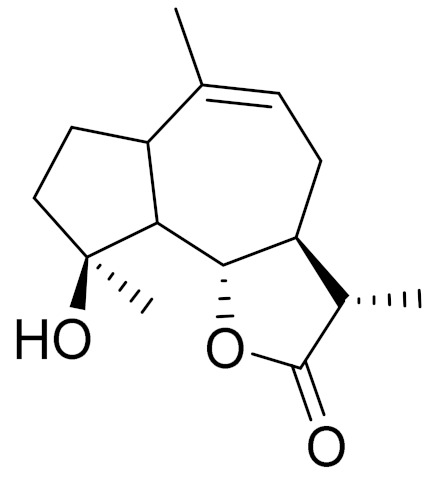
C_15_H_22_O_3_Artaucin [143,144]-
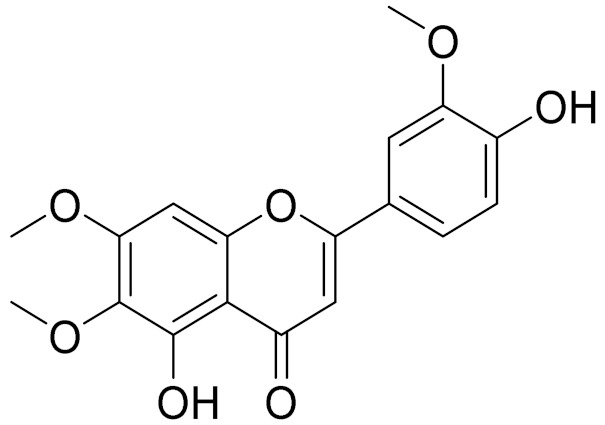
C_18_H_16_O_7_Cirsilineol [112]Antioxidant, cytostatic,antimicrobial, antifungal, antimalarial and antileishmaniasis activity [112]
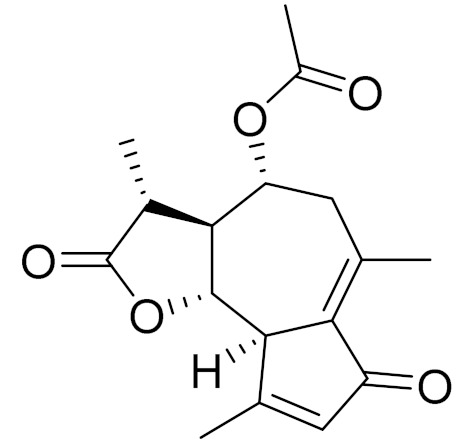
C_17_H_20_O_5_Matricarin-
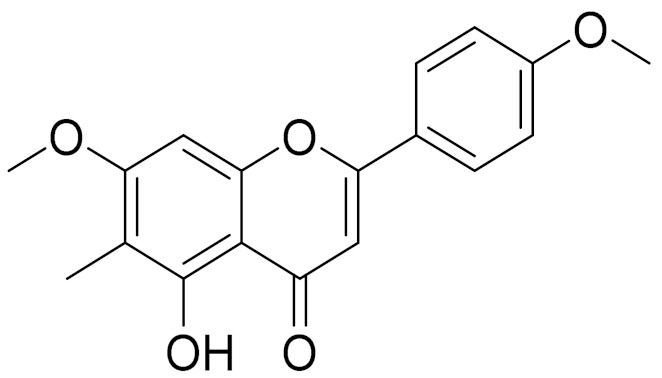
C_18_H_16_O_7_5-oxy-7,4′-dimethoxy-6-methylflavone[144]-
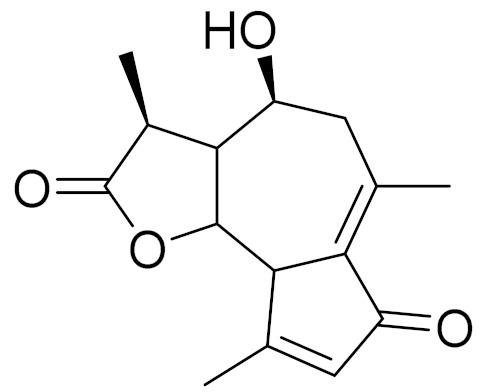
C_15_H_18_O_4_Austricin [143]Angioprotector and antilipidemic activity [126]
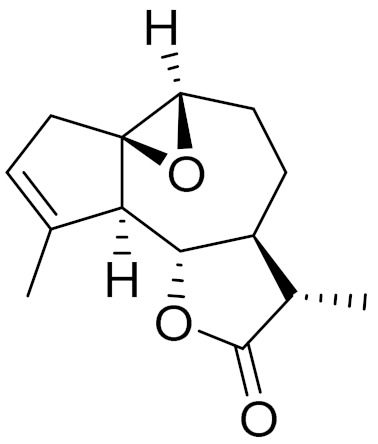
C_14_H_18_O_3_Arborescin [143]Significant cytotoxic activity in vitro [145]*21**Artemisia latifolia*Ledeb.
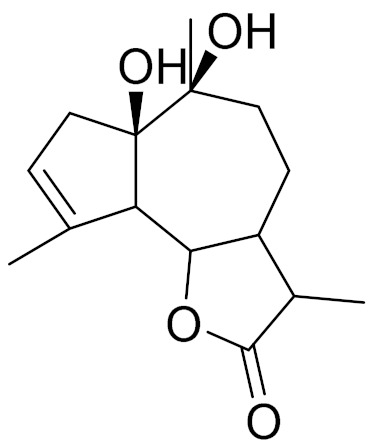
C_15_H_22_O_4_Arlatin [146]-*22**Artemisia sublessingiana* Krasch. ex Poljakov
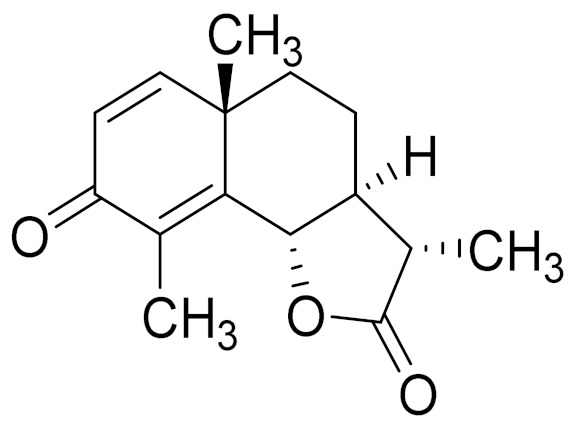
C_15_H_18_O_3_*α*-Santonin [63]Anthelmintic [63] andantipyretic activity [104]
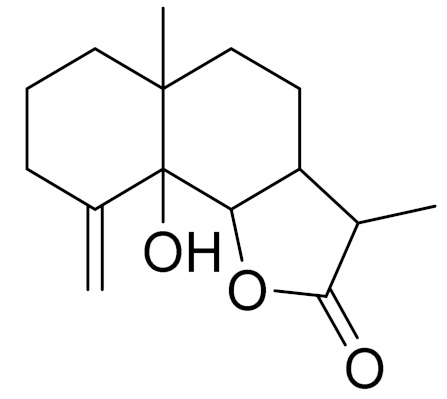
C_15_H_22_O_3_Arsubin [95]Slightly shows antipyretic actions [96]
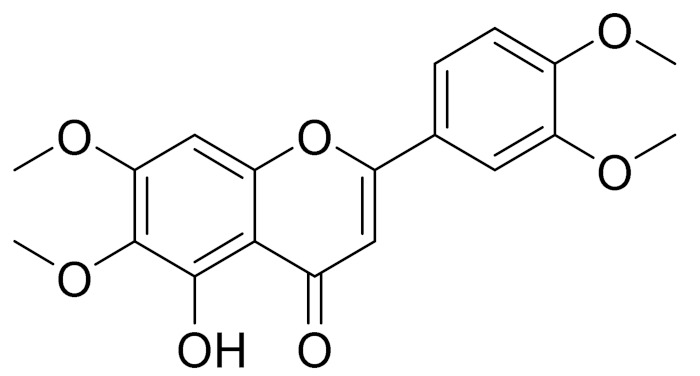
C_19_H_18_O_3_Eupatilin [95,96]Anti-inflammatory activity [118]
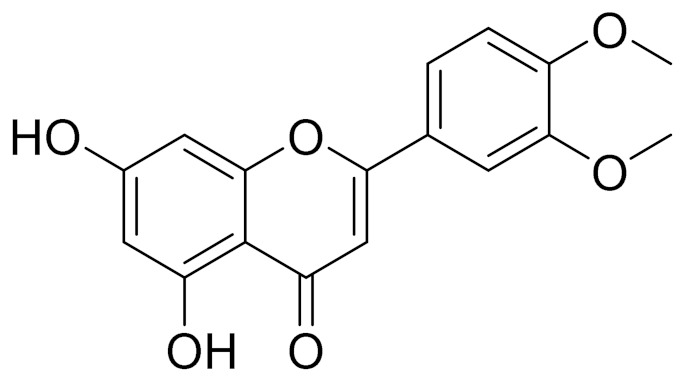
C_17_H_18_O_7_3′,4′-Dimethoxy-luteolin [95,96]Potential against the contagious virus SARS-CoV-2 [98]
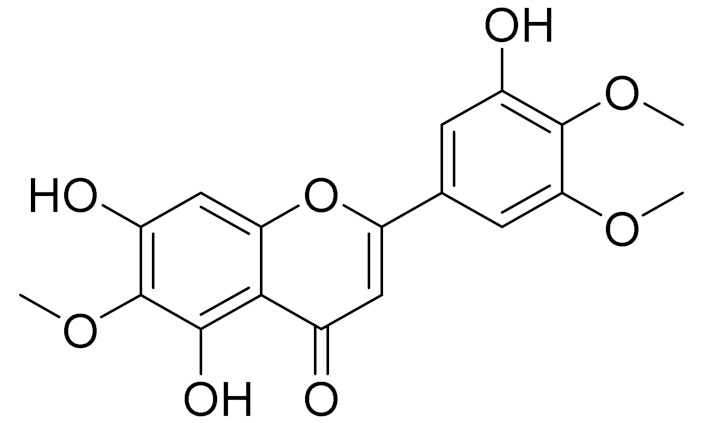
C_18_H_16_O_8_5, 7, 3′-trihydroxy-6,4′,5′-trimethoxyflavone-
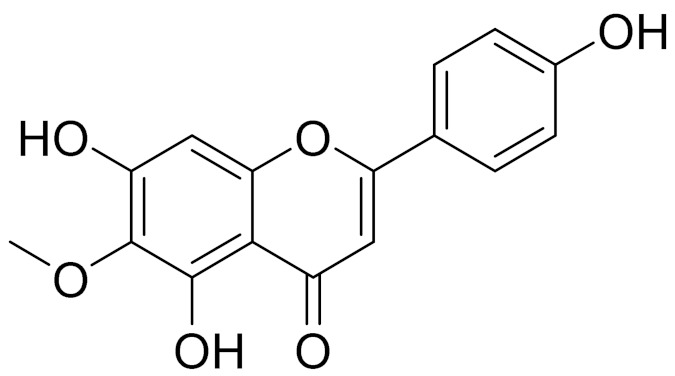
C_16_H_12_O_6_HispidulinAnti-tumor effects in a wide array of human cancer cells [147]
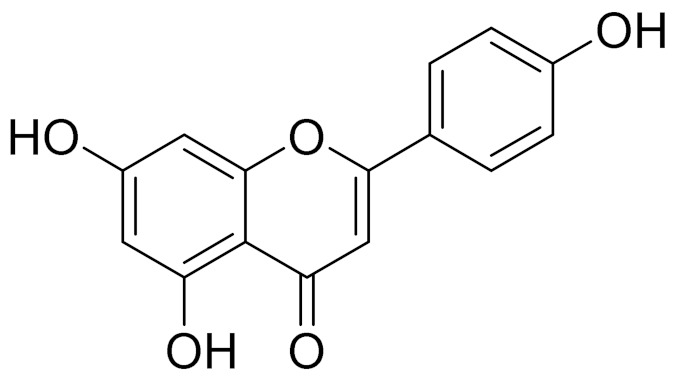
C_15_H_10_O_5_ApigeninAnti-inflammatory, antibacterial, antiviral and antioxidant agent. [148]
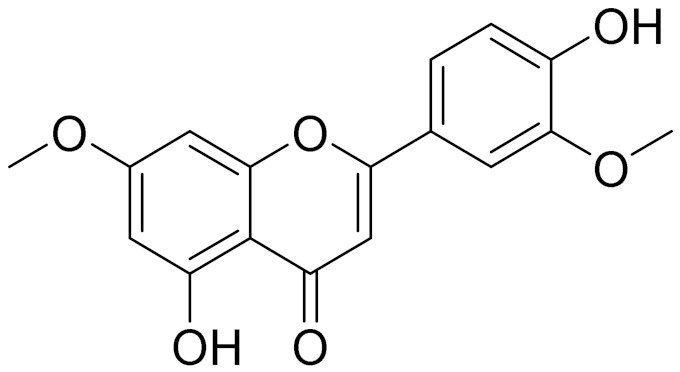
C_17_H_14_O_6_VelutinShows improved inhibitory activity against melanin biosynthesis [149]
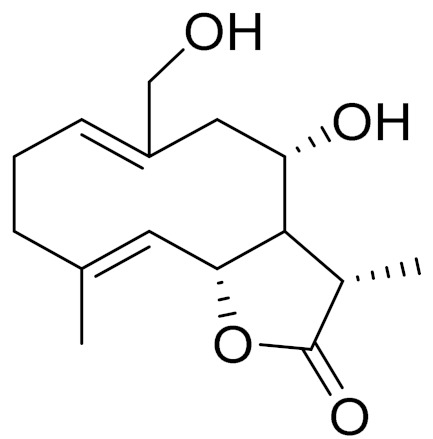
C_15_H_22_O_4_8*α*,14-dihydroxy-11,13- dihydromelampolide-*23**Artemisia nitrosa* Weber
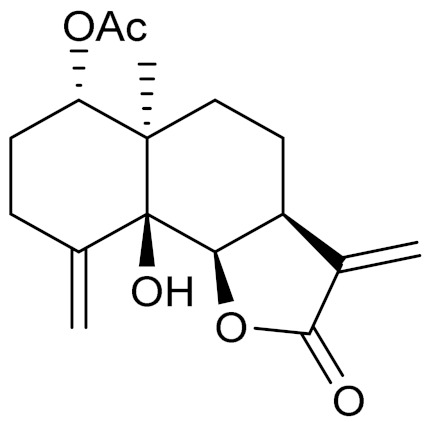
C_17_H_22_O_5_Nitrosin [150]-
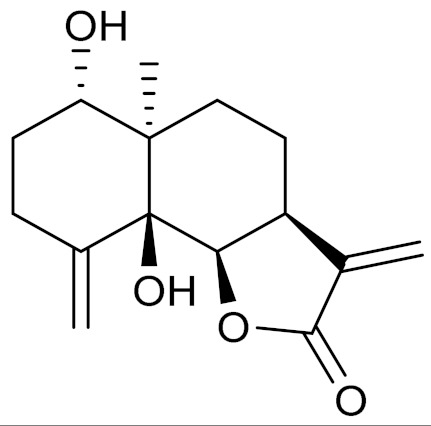
C_15_H_20_O_4_Artemin [151]Promising candidate for the treatment of neurological disorders [128]
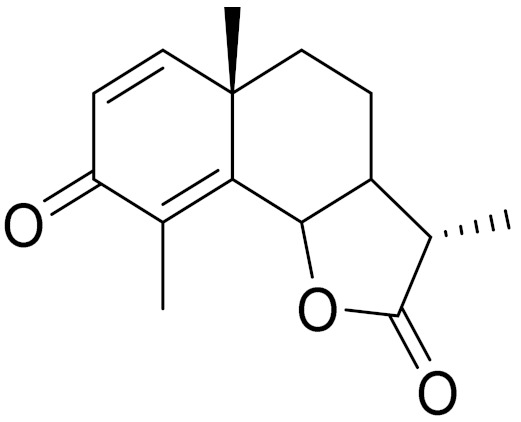
C_15_H_18_O_3_*α*-Santonin [150,151]Anthelmintic [63] andantipyretic activity [104]*24**Artemisia pauciflora* Weber
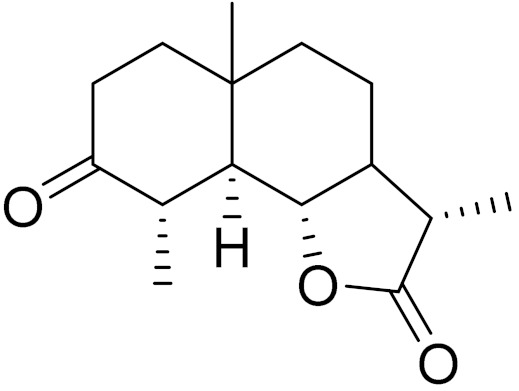
C_15_H_22_O_3_3-oxo-5,7a,4,6,11b(H)-eudesman-6,12-olide [152]-*25**Artemisia transiliensis* Poljakov and *Artemisia serotina* Bunge
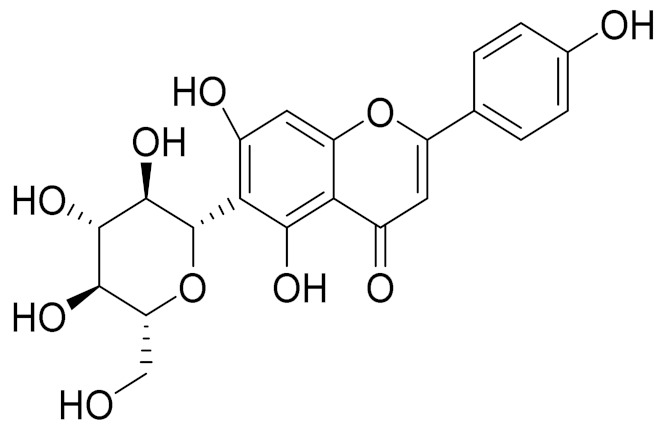
C_21_H_20_O_10_Isovitexin [90]Antidiabetic agent [153]
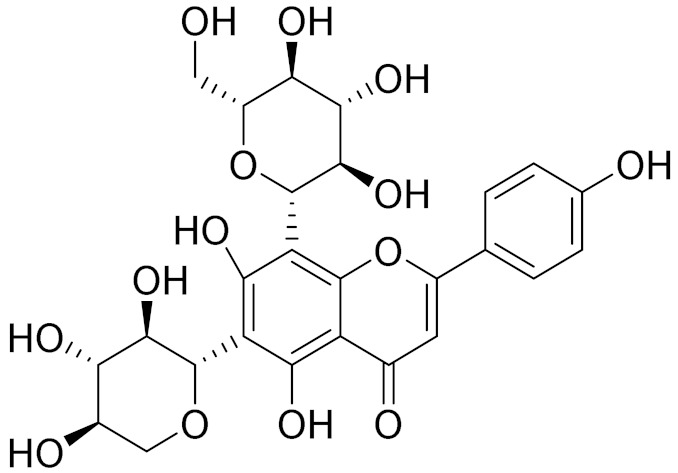
C_26_H_28_O_11_Vicenin 1Inhibitory effect on angiotensin-converting enzymes [154]
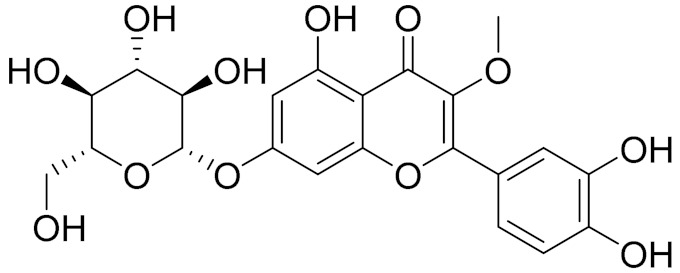
C_22_H_22_O_12_Vransilin-
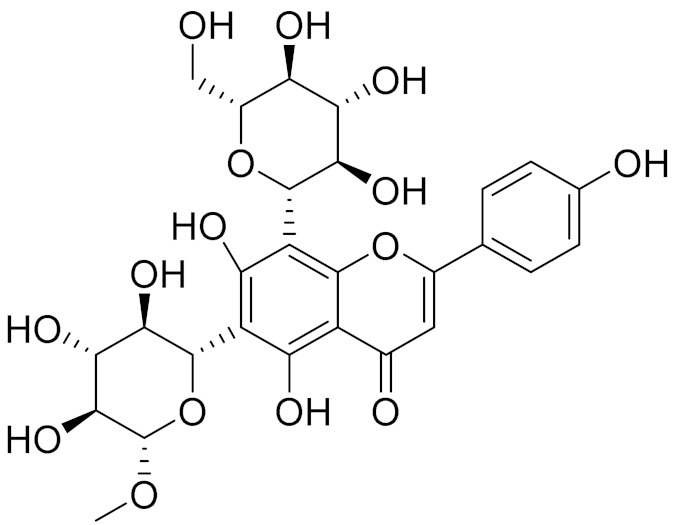
C_27_H_30_O_15_Vicenin 2Anti-inflammatory activity [155]
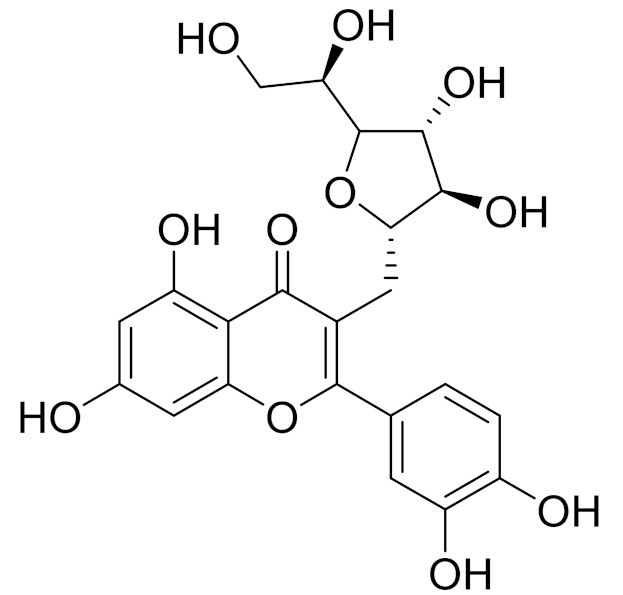
C_22_H_22_O_11_IsoquercitrinChemoprotective effects, both in vitro and in vivo, against oxidative stress, cancer, cardiovascular disorders, diabetes and allergic reactions [156]
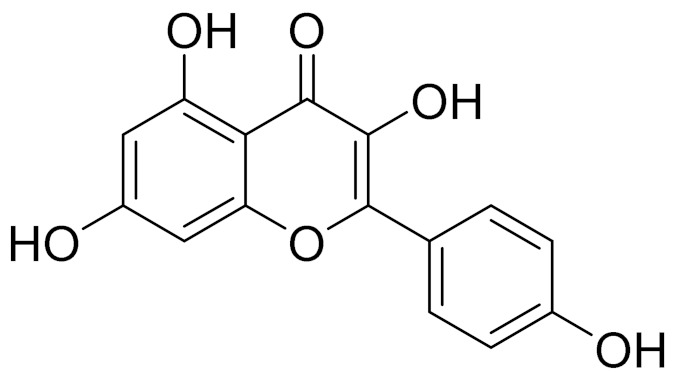
C_16_H_12_O_7_3-*O*-MethylquercetinPossesses antioxidant, antiviral and anticancer properties [157]
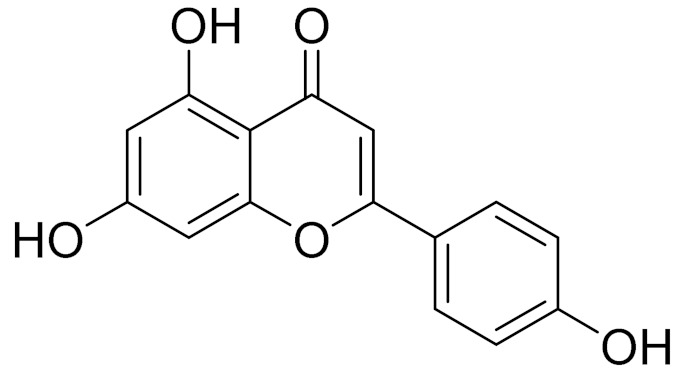
C_15_H_10_O_5_ApigeninAntioxidant, anti-inflammatory and chemoprevention activity [148]
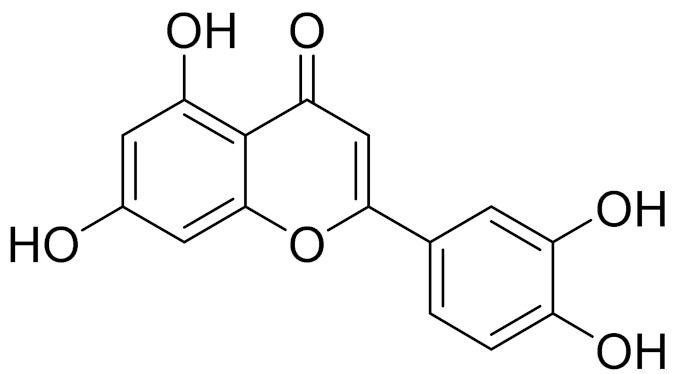
C_15_H_10_O_6_LuteolinAnticancer, anti-inflammatory, antioxidant, anti-allergic and antimicrobial activity [158,159]
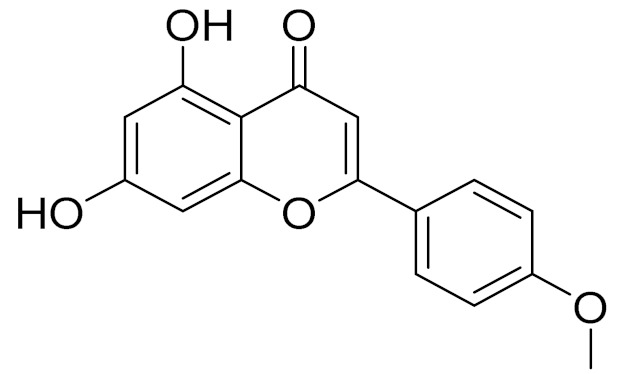
C_16_H_12_O_5_AcacetinAnticonvulsant [160]
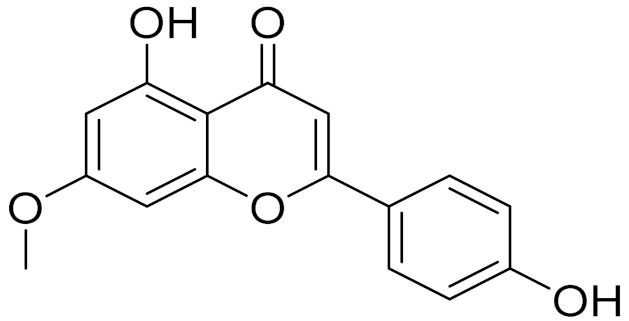
C_16_H_12_O_5_GenkwaninAnti-inflammatory activity [133]
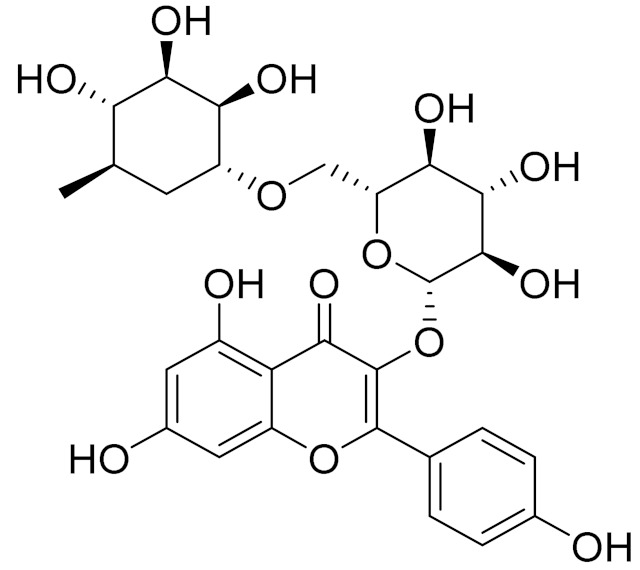
C_28_H_32_O_14_RutinAntimicrobial, antifungal and anti-allergic agent [161]
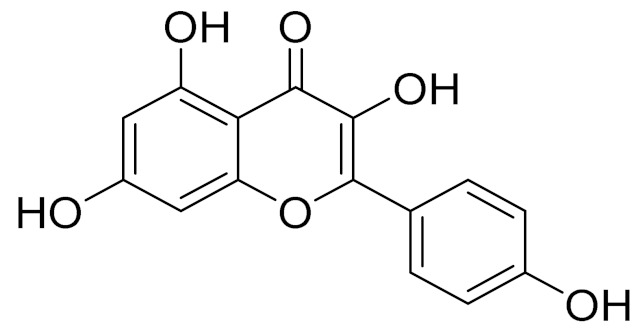
C_15_H_10_O_6_QuercitinPossesses antioxidant properties and is used in the protection against various diseases such as osteoporosis, lung cancer and cardiovascular disease [162]*26**Artemisia commutata* Besser
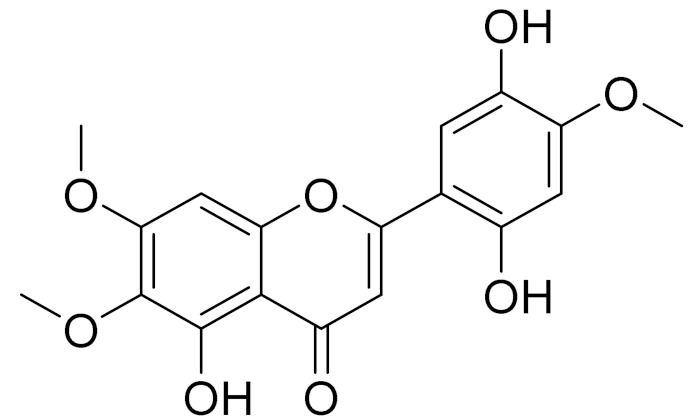
C_18_H_16_O_8_Jusanin [99]Jusanin showed a high structural similarity degree with X77, the co-crystallized legend of the COVID-19 main protease (PDB ID: 6W 63), Mpro. [97]
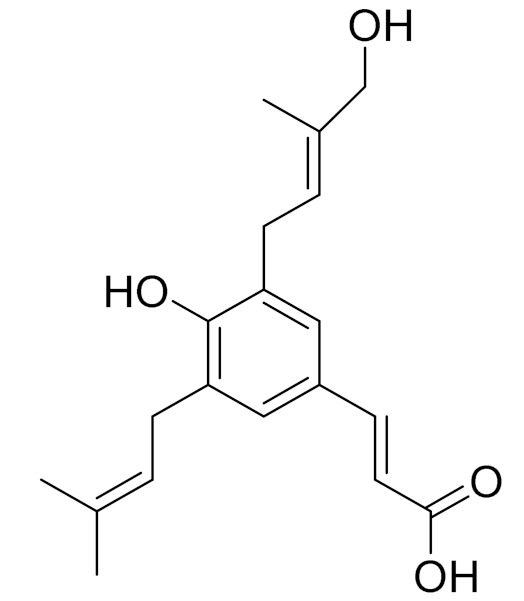
C_19_H_24_O_4_Capillartemisin ACholeretic activity [163]
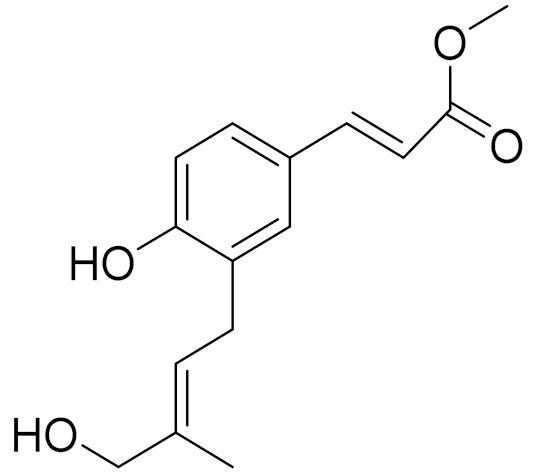
C_15_H_18_O_4_Methyl-3-[S-hydroxyprenyl]-cumarate-
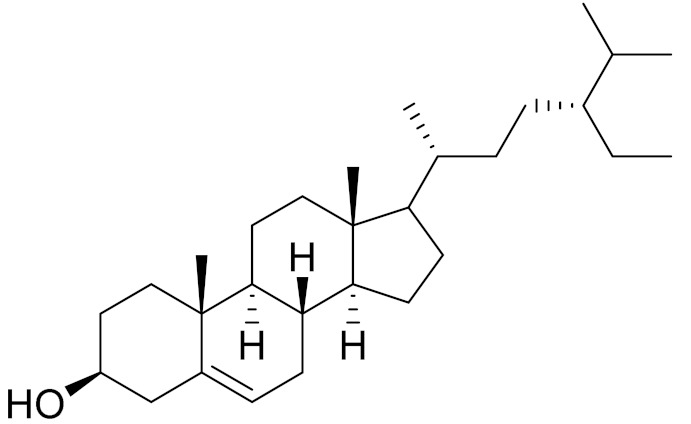
C_29_H_50_O*β*-sitosterolAntifibrotic activity [164]*27**Artemisia glauca* Pall. ex Willd
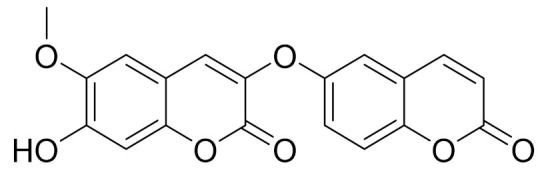
C_19_H_12_O_7_Jusan coumarin [165]Jusan coumarin demonstrated a high degree of similarity with X77, the co-crystallized ligand of Mpro. [98]
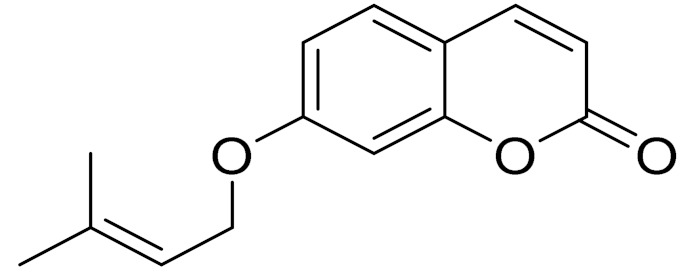
C_14_H_14_O_3_7-isopentenyloxycoumarinAntitumor activity [166]
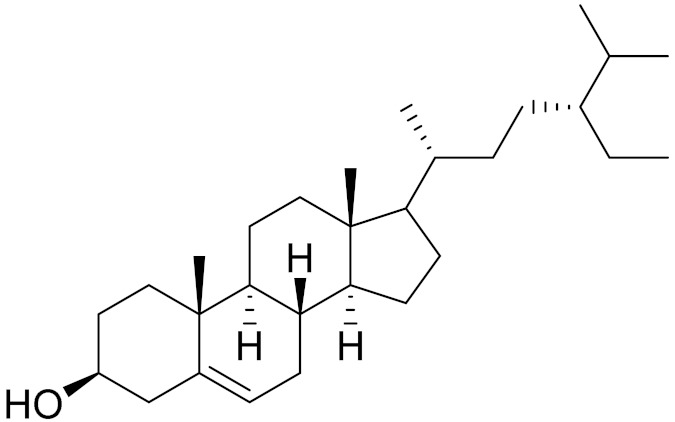
C_29_H_50_Oβ-sitosterolAntifibrotic activity [164]*29**Artemisia santolinifolia* Turcz. ex Bess.
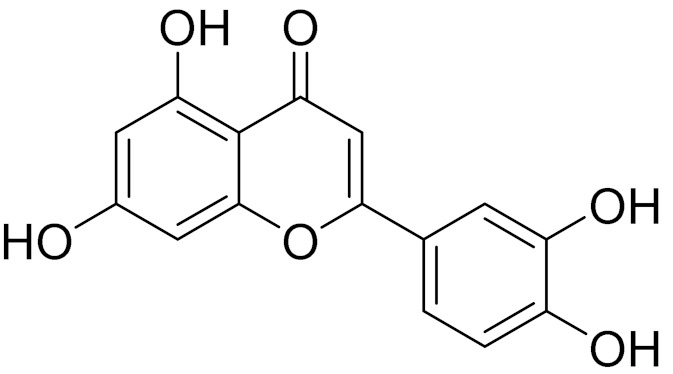
C_15_H_10_O_6_LuteolinAnticancer, anti-inflammatory, antioxidant, anti-allergic and antimicrobial activity [158,159,160,167]
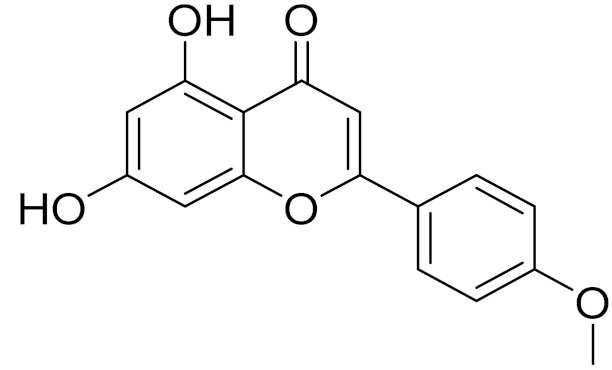
C_16_H_12_O_5_AcacetinAnticonvulsant [160]
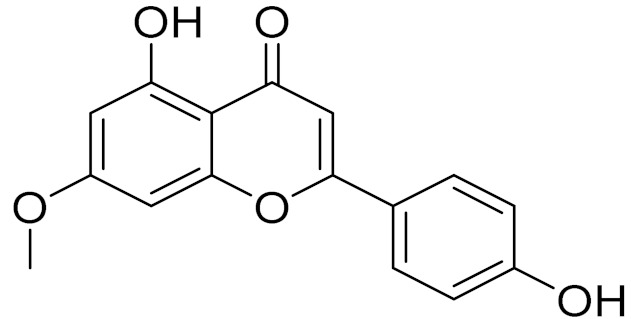
C_16_H_12_O_5_GenkwaninAnti-inflammatory activity [133]
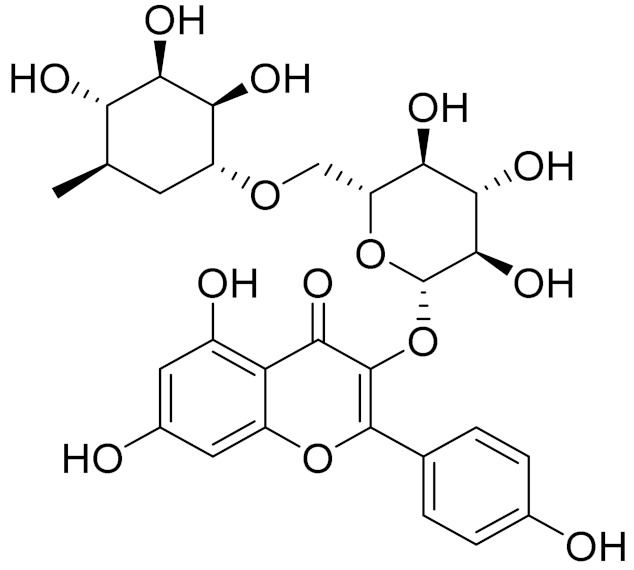
C_28_H_32_O_14_RutinAntimicrobial, antifungal and anti-allergic agent [161]
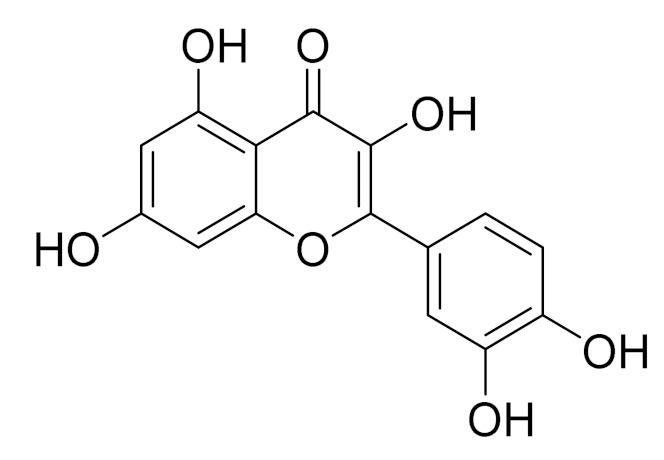
C_15_H_10_O_6_QuercitinPossesses antioxidant properties and is used in the protection against various diseases such as osteoporosis, lung cancer and cardiovascular disease [162]
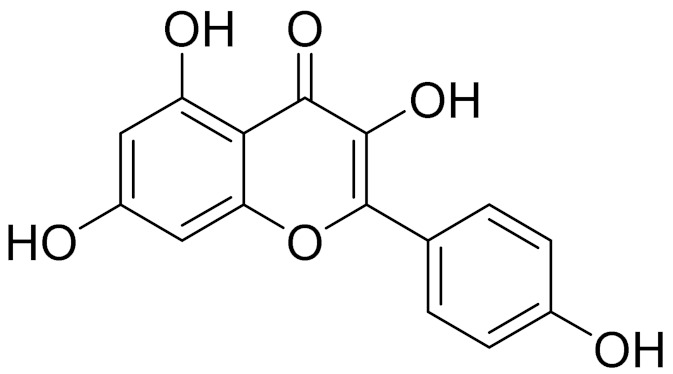
C_15_H_10_O_6_KaempferolAntioxidant and antibacterial agent, as well as a plant metabolite [168]
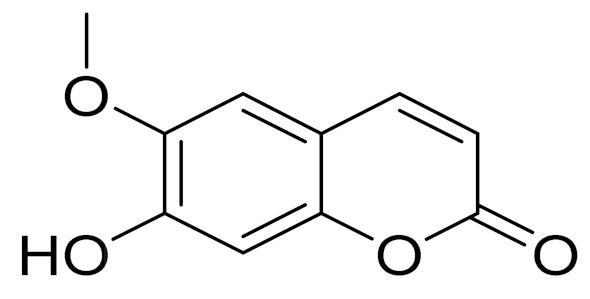
C_10_H_8_O_4_ScopoletinPotential antineoplastic, antidopaminergic, antioxidant, anti-inflammatory and anticholinesterase effects [169]
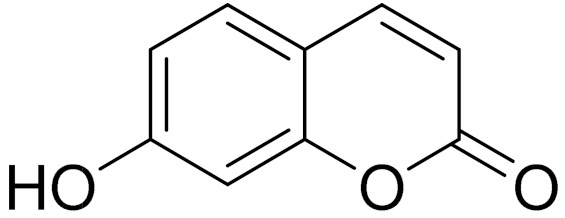
C_9_H_6_O_3_UmbelliferoneAntioxidant properties [170]*30**Artemisia aralensis* Krasch.
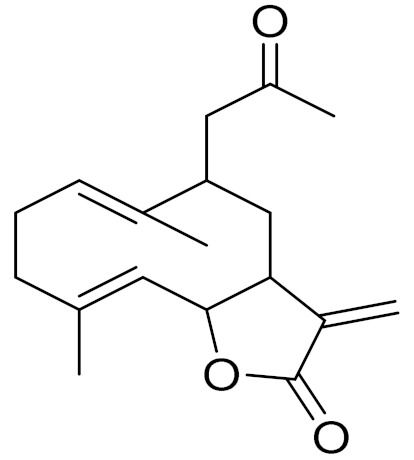
C_18_H_24_O_3_Argracin [91]TCR activity [139](-) means not studied.


## 4. Essential Oil Contents of *Artemisia* Species from Central Asia

*Artemisia* species have a strong odor due to the availability of essential oils [99]. The major compounds of the essential oils of *Artemisia* species from Kazakhstan have been summarized in Table 2, according to the 10 published articles up to now. The differences in the quantity or quality of the compounds may be attributed to the change in the pH of the soils, the geographic location, the climate, the chemotypes, the subspecies, the collection time, the drying conditions and the extraction methods.

Camphor, 1, 8-cineole and thujones have been identified as the major compounds in the essential oils of *Artemisia* species growing in Kazakhstan. Camphor (Figure 3, Table 2) is a cyclic monoterpene ketone with a strong mothball smell. It has been reported that camphor has a wide range of biological activities, such as antiviral, antimicrobial, antitussive and analgesic agent activities [165]. 1,8-cineol (Figure 3, Table 2) is an aromatic component which is used for the treatment of respiratory tract diseases due to its antimicrobial, mucolytic, broncholytic and anti-inflammatory properties. α,β-thujones (Figure 3, Table 2) are volatile monoterpene ketones with antimalarial, antiviral, antitumor, spasmolytic and other effects [65]. The characteristics of the essential oils that are already seen in nature have mostly been used. They are used in food preservation, fortification and as microbicidal, analgesic, sedative, anti-inflammatory, spasmolytic and locally anesthetic treatments. They are well known for their antiseptic (bactericidal, virucidal and fungicidal) and therapeutic capabilities, as well as their scent [66]. Essential oils, or some of their components, are utilized in sanitary products, dentistry, agriculture, fragrances and cosmetics, food additives and preservatives, agrochemicals and natural medicines. Except for the fact that more is understood about some of their modes of action, particularly at the antimicrobial level, these traits have remained substantially constant up to the present. Additionally, essential oils have the power to prevent bacterial cells from synthesizing proteins, polysaccharides, DNA and RNA.

The essential oil of *A. terrae-albae* has high contents of camphor (47.3%), 1,8-cineole (23.9%), camphene (9.8%) and β-thujone (6.0%). According to the literature, the essential oil of *A. glabella* contains mostly 1,8-cineole (12%), linalool (8%), terpine-4-ol (6.5%), α -terpineol (5%) and sabinene derivatives (up to 5%) [75]. *A. frigida* contains 1,8-cineole (24.7%), camphor (22.6%) and borneol (8.9%), where thujone (5.2%) and thujanols (1.3–2.5%) were distinguished quantitatively [80]. *A. scopaeformis* hexane extract contains four main compounds: methyleugenol (33.87%), hexadecanoic acid, ethyl ester (41.02%), butyl 4,7,10,13,16,19-docosahexaenoate (11.55%) and hexasiloxane (13.56%) [89,90]. The chloroform extract of *A. scopaeformis* contains five major compounds: fluorene, 2,7-bis (1-hydroxyethyl 24.28%), p-imethylaminobenzylidene p-anisidine (14.59%), 3-acetoxy-5-methyl-2-nitro- terephthalic acid, 4-isopropyl ester 1-methyl ester (11.74%), 3,4-diacetyl-2-methyl-4H-thieno [3,2-b] pyrrole-5-carboxylic acid, methyl ester (17.35%) and 4H-1,2,4-triazole-3-thiol, 4-2-fluorophenyl)-5-(1-methylethyl) (32.04%) [89,100]. The major components of *A. rupestris* essential oil were myrcene (9.5%), *β*-elemene (5.4%) and capric acid (5.1%). The oil of *A. glabella* was found to be rich with 1,8-cineole (12.2%), cumin aldehyde (9.4%), α-terpineol (5.7%) and borneol (5.2%) [70]. The principal components in *A. sieversiana* were myrcene, (14.2%) 1,8-cineol (9.3%), linalool (4.2%), *p*-cymene (3.4%), nerylisovalerate (3.4%) and caryophyllene (3.0%). The major components in *A. lercheana* were β-thujone (45.6%), α-thujone (24.2%), camphor (7.5%) and 1,8-cineol (4.6%) [69].

The research data indicates that the essential oil from the *Artemisia* genus of Kazakhstan has antibacterial, antifungal and antiviral activities. The study of the antimicrobial activity of the essential oil from *A. sieversiana* showed that it was active toward gram-negative strains and yeasts [69]. The determination of the major components of essential oils from *Artemisia* species growing in Kazakhstan showed that 1,8-cineole, α-thujone, β-thujone and camphor were present in higher amounts among most of the plant species (Table 2).

## 5. Chemical Constituents and Bioactivity Ascertainment of *Artemisia* L. Species from Central Asia

### 5.1. Materials and Methods

#### 5.1.1. Instruments and Chemicals

Enzymatic experiments, Total Phenolic Content (TPC), Total Flavonoid Content (TFC) and radical scavenging assays (RSA) were carried out on a SpectraMax M3 Multi-Mode Microplate Reader (Molecular device, USA). α-Glucosidase (EC3.2.1.20), BNA (EC 3.2.1.18), 2,2-Diphenyl-1-picrylhydrazyl (DPPH), 2,2’-azino-bis(3-ethylbenzothiazoline-6-sulfonic acid) (ABTS), Folin-Ciocalteu reagent, gallic acid, 6-hydroxyl-2,5,7,8-tetramethylchroman-2-carboxylic acid (Trolox), dimethyl sulfoxide (DMSO) and other chemicals used for the assays were purchased from Sigma Aldrich (St. Louis, MO, USA). The PTP1B (EC 3.1.3.48, human, recombinant) enzyme was bought from Enzo Life Science.

#### 5.1.2. Preparation of Extracts for Enzymatic Assays

The dried species (1 g) of *Artemisia* were extracted using methanol (50 mL) at room temperature to obtain the crude extract.

#### 5.1.3. α-Glucosidase Inhibitory Activity Assay

The inhibitory activity of α-glucosidase was carried out with a few changes from the method reported in the literature [171], using *p*-nitrophenyl- α-_D_-glucopyranoside (*p*-NPG) at the optimal pH of 6.8 (50 mM phosphate buffer). Extracts were dissolved and diluted to a needed concentration in DMSO. Concisely, in 96-well plates, 10 μL of extracts or deoxynojirimycin (DNJ) as a control and 40 μL substrate (*p*-NPG, 1.0 mM) in the aforesaid buffer (130 μL) were added 20 μL of the enzyme (0.1 unit/mL). The absorbance of formed *p*-nitrophenol immediately measured with a wavelength of 405 nm at 37 °C. The compounds activity was expressed in the concentration when 50% of the enzyme activity was inhibited (IC_50_). The calculation of the % of inhibition was as follows:Activity (%) = 100 [1 + ([I]/IC_50_)](1)

#### 5.1.4. Assay of PTP1B Inhibitory Activity

The PTP1B inhibitory activity of extracts was measured according to the previously published research work [172]. The extracts and positive control were solubilized in DMSO, and the needed concentration was achieved through dilution. The Tris-HCl (pH 7.5) buffer was prepared by taking 25 mM Tris, 1 mM ethylenediaminetetraacetic acid (EDTA), 2 mM 2-mercaptoethanol and 1 mM dithiothreitol, and the pH was achieved using HCl. The following reaction was performed in a 96-well plate: 130 µL buffer, 10 µL of the sample and 40 µL of *p*-nitrophenyl phosphate (*p*NPP, 0.8 mM treated concentration) as a substrate, and the last 20 µL of the enzyme (1 µg/mL treated concentration) were put and incubated for 10 min at 37 °C. The reaction of the subsequent hydrolysis of *p*NPP was monitored for 30 min at 405 nm. The half-maximal inhibitory concentration (IC_50_) was validated from the transformation of Equation (1).

#### 5.1.5. BNA Inhibition Assay

For performing the BNA inhibition assay, the measurement of fluorescence was done according to previously published methods [173]. The emission wavelength was 450 nm, the excitation wavelength was 365 nm and the reaction was performed in a 96-well black immuno-microplate (SPL Life Sciences, Korea) at 37 °C. First, 20 µL of 1 mM of the substrate (4-methylumbelliferyl-*N*-acetyl-*α*-_D_-neuraminic acid sodium salt) aqueous solution was mixed with 160 µL of 50 mM sodium acetate buffer (pH 5.0). Then, 10 µL of the testing solution and 10 µL of enzyme (0.2 units/mL) were immediately added. The inhibitor concentration leading to a 50% loss in enzyme activity (IC_50_) was obtained from Equation (1).

#### 5.1.6. Determination of TPC

The TPC of *Artemisia* species was determined according to the Folin–Ciocalteu assay on the basis of the calibration curve plotted using gallic acid diluted in DMSO (0–500 µg/mL) [174]. Methanol extracts of plants (40 µL) were added to the 1.5 mL Eppendorf tube, in which 40 µL of Folin–Ciocalteu reagent was diluted in 360 µL of distilled water (DW) and incubated for 5 min. Then, the 7% *w*/*v* solution of 400 µL of sodium carbonate in 160 µL of DW was added and again incubated for 90 min in a dark place. After incubation, the mixture was centrifuged at 13,000 rpm for 5 min, and the absorbance of 200 µL of the supernatant was measured at 750 nm using a 96-well plate. The TPC was indicated as mg of gallic acid equivalents per 100 g of the sample (mg GAE/100 g).

#### 5.1.7. Determination of TFC

To determine the TFC, a calibration curve drawn from different concentrations of quercetin diluted in DMSO (0–500 µg/mL) was used [175]. First, in a 1.5 mL Eppendorf tube, extracts (40 µL) of 200 µL DW with 15 µL 5% *w*/*v* sodium nitrite solution were added and incubated for 5 min. Then, 15 µL of 10% *w*/*v* aluminum chloride solution was added thereto and incubated for 6 min. After this, 100 µL of 1 M sodium hydroxide in 120 µL of DW were added to the reaction and mixed well. A total of 200 µL of the resulted mixture was placed on the 96-well plate, and its absorbance was recorded at 415 nm. The TFC was expressed as mg of quercetin equivalents per 100 g of the sample (mg QE/100 g).

#### 5.1.8. DPPH Radical Scavenging Assay

The RSA on the DPPH radical was evaluated according to Brand-Williams et al. (1995), adapted to 96-well microplates [174] with modifications. The samples (10 μL, in different concentrations in M) were mixed with 190 μL of DPPH solution methanol in 96-well flat bottom microplates and incubated in the darkness at RT for 10–15 min. The absorbance was measured at 517 nm, and RSA was expressed as the percentage of inhibition relative to a control containing methanol in place of the sample and as the half-maximal inhibitory concentration (IC_50_, μM). Trolox was used as a positive control.
Radical scavenging activity (%) = [(I_0_ − I)/I_0_] × 100(2)

#### 5.1.9. ABTS Radical Scavenging Activity

The RSA on the ABTS^•^^+^ radical was evaluated by the procedure described in [175]. A stock solution of ABTS^•^^+^ (7 mM) was prepared by mixing equal amounts of ABTS and potassium persulfate (2.45 mM) at room temperature in the dark for 14 h. The ABTS^•+^solution was diluted with ethanol to obtain an absorbance of at least 0.7 at 734 nm. The samples (10 µL at different concentrations in µM) were mixed with 190 µL of ABTS^•^^+^solution in a 96-well microplate. After incubation for 1 min, the absorbance was measured. The results are expressed as the inhibition concentration (IC_50_) of 50% radical scavenging. Trolox was used as a positive control.

### 5.2. Results and Discussion

Protein Tyrosine Phosphatase 1B (PTP1B, EC 3.1.3.48) and α-glucosidase (EC 3.2.1.20) are the most crucial enzymes for diabetes mellitus, which is a chronic disorder evoked by the high level of blood sugar. The PTP1B is vastly expressed in tissues such as fat, muscle and liver. PTP1B appears as a key regulator of insulin-receptor activity that acts at the insulin receptor and downstream signaling components, such as the insulin receptor substrate. Moreover, PTP1B levels also seem to be raised in particular physiological or pathophysiological settings of leptin resistance, which is linked to food uptake, causing obesity. The α-Glucosidase enzyme is found in the small intestine and catalyzes the breakdown of sugar into glucose. Abnormal amounts of the α-glucosidase enzyme lead to severe blood sugar-related illnesses such as diabetes. Oxidative stress leads to the accumulation of free radicals including reactive oxygen species (ROS), which can seriously damage cell components (lipids, proteins and DNA), and it is suggested to be a trigger for many pathological factors such as cancer, asthma and diabetes. Another enzyme, bacterial neuraminidase (BNA), is the virulence factor of many pathogens, bacteria and viruses. The BNA (EC 3.2.1.18) is from the group of exo-sialidases which cleaves the α-ketosidic bond connecting the terminal sialic acid residue with the adjoining oligosaccharide fragment. Sialic acid linkage is very necessary for infections by pathogenic bacteria such as *Clostridium perfrigens*. The enhancement of the adhesion of *C. perfringens* is due to the negative charges of sialic acids and their ability to disrupt the integrity of the endothelial barrier. Thus, neuraminidase inhibitors could be involved in the infection step because sialic acid linkage is one of the target recognition points for bacteria. With the help of neuraminidase, bacteria multiply from cell to cell, since the enzyme speeds up the reaction of pinching off bacteria from the first cells which the pathogen managed to infect.

Biological activity to extinguish the catalytic activity of enzymes with *Artemisia* methanolic extracts was tested using a SpectraMaxM3 spectrophotometer according to previously published methods. As a result, extracts (*Artemisia*) showed potential activity to inhibit the enzymes α-glucosidase, PTP1B and BNA and antioxidant activity. Among them, *A. scopaeformis*, *A. albicerata, A. transiliensis*, *A. schrenkiana* and *A. albida* showed a higher-than-50% inhibition of α-glucosidase at a concentration of 50 μg/mL. Similarly, this species also showed the highest activities with the PTP1B enzyme at a concentration of 50 μg/mL. Moreover, all species were significantly potent against BNA even at a lower concentration of 20 μg/mL. The tested *Artemisia* species and the percentage of inhibition at a concentration of 50 μg/mL are given in Table 4.

TPC, TFC, DPPH and ABTS RSA were initially quantified with methanol extracts. The high potencies of the methanol extract recorded on *A. schrenkiana*, *A. scopaeformis*, *A. transiliensis* and *A. scoparia* were 5199, 5804, 4166 and 4711 mg GAE/100 g of TPC, respectively. The TFC values were high on similar species: 2080, 2745, 2975 and 1951 mg QE/100 g, respectively (Figure 4). The *A. rutifolia* also showed a higher content of TFC than the others (2024 mg QE/100g). The RSA (Figure 5 and Figure 6) of these species were correlated to TPC and TFC, which is shown in Table 5 and Figure 4. These results indicate that methanol extracts have antioxidant potentials according to their phenolic and flavonoids contents and their ability to scavenge reactive radicals (ABTS and DPPH) in comparison with the positive control (trolox).

## 6. Conclusions

The interest in alternative medicine has always held a special place in human history. The treatment of diseases with the help of various medicinal herbs, including plants of the *Artemisia* species, has reached a new level. The worldwide interest in herbal medicines is currently growing, followed by increased laboratory investigations into the pharmacological properties of the bioactive ingredients and their potential to treat different diseases. By studying ethnopharmacology and conventional medicine, many medicines have entered the market. Thus, it was explained that the by-products of plant metabolism, such as terpenoids, monoterpenes in essential oils, sesquiterpene lactones, flavonoids, isoprenoids and alkaloids, are responsible for biological activities, including antibacterial, antimalarial, antiviral, anti-inflammatory, anticancer, antiplasmodial, antiepileptic and other activities.

This review provides an overview of *Artemisia* species from Central Asia, particularly traditional uses in folk medicine and the recent numerous phytochemical and pharmacological studies. Furthermore, our aim was to search for promising, potentially active *Artemisia* species candidates, encouraging us to analyze PTP1B, α-glucosidase and BNA inhibition as well as the antioxidant potentials of *Artemisia* plant extracts, in which endemic species have not been explored for their secondary metabolites and biological activities so far. Among all the species, *A. scopaeformis*, *A. albicerata, A. transiliensis*, *A. nitrosa, A. schrenkiana* and *A. albida* showed a high potential for α-glucosidase, PTP1B and BNA inhibition, which is associated with diabetes, obesity and bacterial infections. The antioxidant potentials of the species *A. schrenkiana*, *A. scopaeformis*, *A. transiliensis* and *A. scoparia* were also promising. Thus, our results contribute to the human health benefits of *Artemisia* species based on the inhibition of enzyme inhibitions. In general, the phenolic contents were correlated with those of flavonoid and biological activities. The methanol extracts of these *Artemisia* species exhibited considerably high antioxidant activity. The presence of phenolic compounds in our extracts should be the main cause of their high antioxidant power. However, the examination of details between different *Artemisia* species in our research has shown that other species are also good materials for the antioxidant functional natural source. This is the first report indicating that the endemic species *A. scopaeformis*, *A. transiliensis, A. schrenkiana* and *A. albicerata* and the other species *A. nitrosa* and *A. albida* showed biological activities. An extensive review of the literature demonstrates that several *Artemisia* species exhibit a wide range of biological activities, including antimalarial, anticancer and anti-inflammatory actions. More thorough research is required in this area, although there is a significant potential for the bioactive chemicals from Artemisia to offer significant alleviation from a variety of human illnesses.

As a result, formulations based on *Artemisia* may be employed as innovative, secure and affordable medicines or even as antiviral nutraceuticals to increase immunity and provide tolerance to viral infections. Due to their low cost, complex therapeutic effect on the body, low toxicity and potential for long-term usage without adverse effects, the use of medicinal plants has increased recently. It looks very hopeful that this field will advance through the use of medicinal plants in healthcare and the expansion of the variety of phytopreparations.

## Figures and Tables

**Figure 1 molecules-27-05128-f001:**
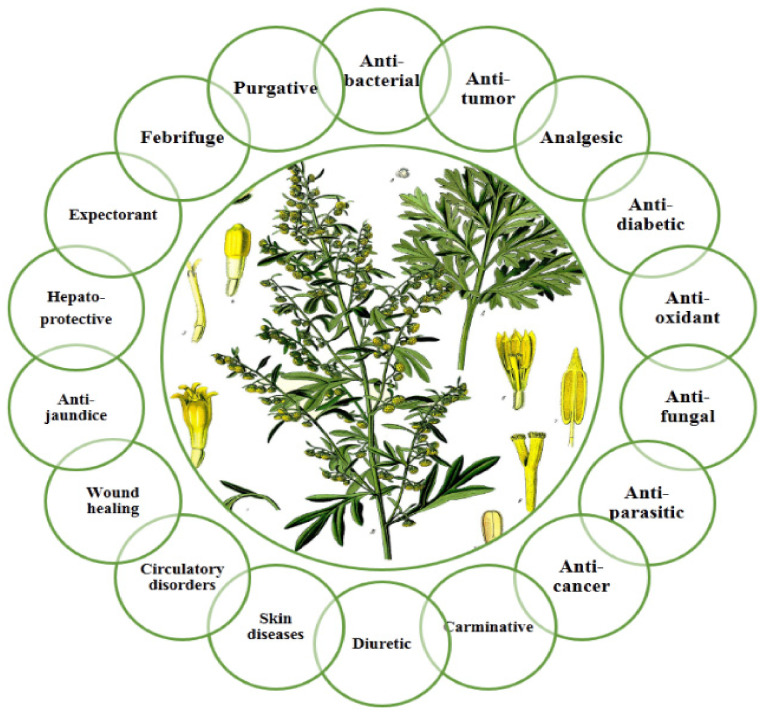
Health benefits of *Artemisia* genus.

**Figure 2 molecules-27-05128-f002:**
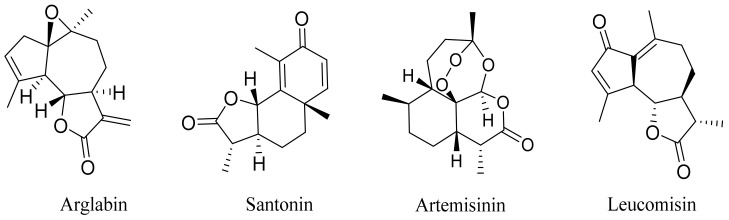
Prominent examples of sesquiterpene lactones isolated from *Artemisa* L.

**Figure 3 molecules-27-05128-f003:**
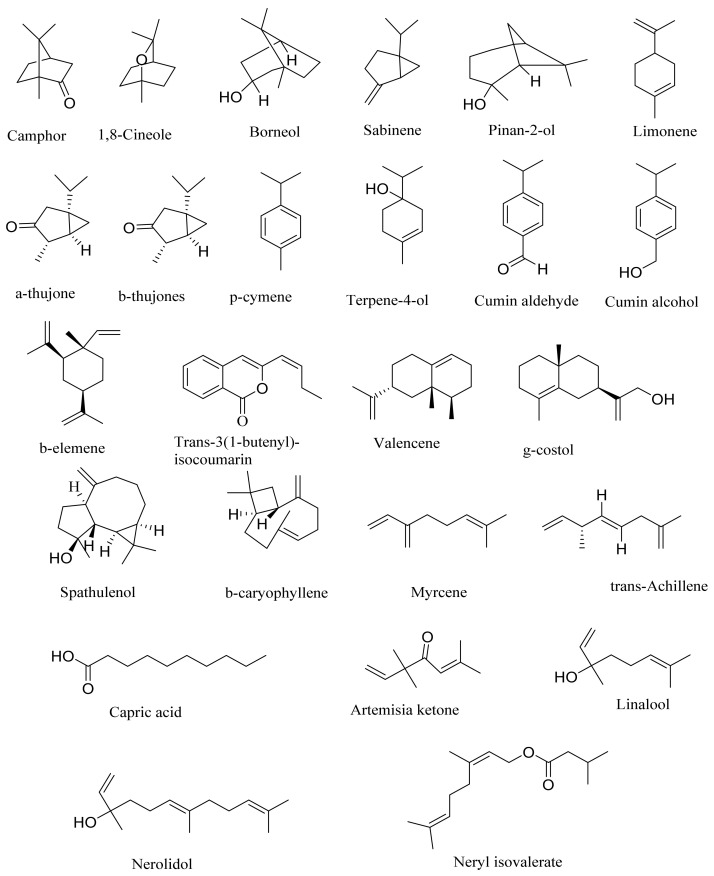
Structures of major components in essential oils from *Artemisia* L.

**Figure 4 molecules-27-05128-f004:**
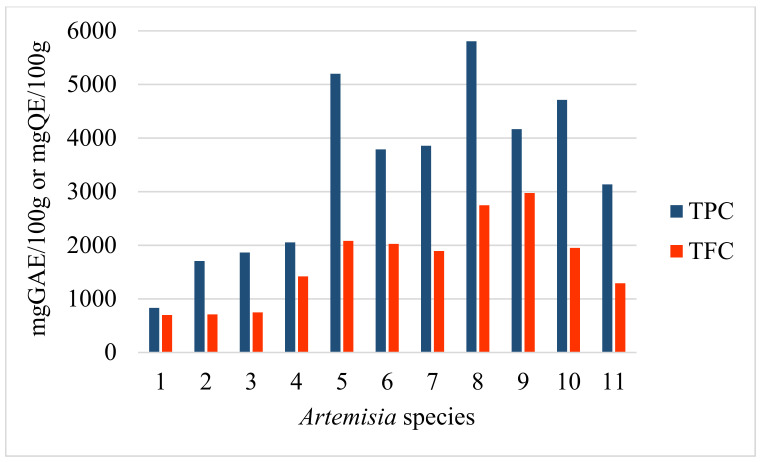
TPC and TFC of *Artemisia* species.

**Figure 5 molecules-27-05128-f005:**
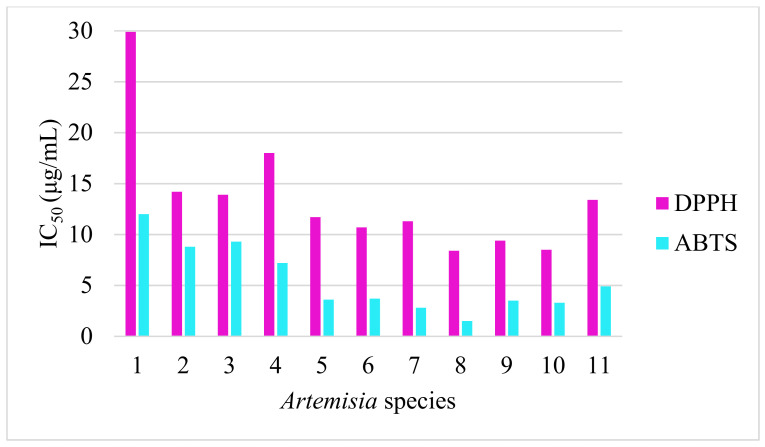
RSA of *Artemisia* species.

**Figure 6 molecules-27-05128-f006:**
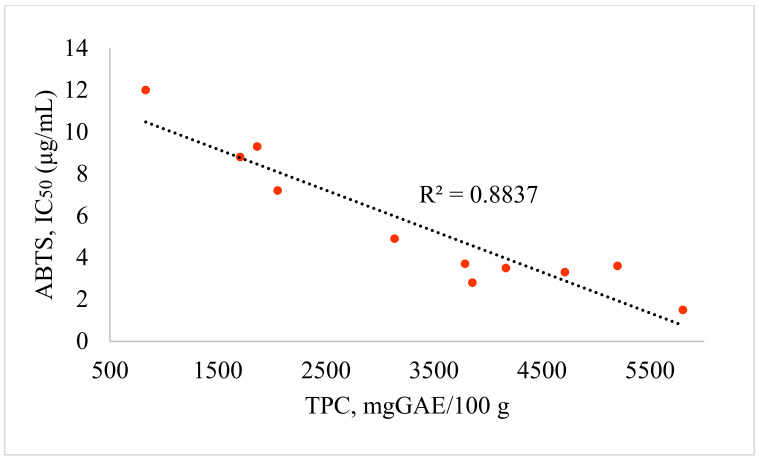
Correlation between the TPC and RSA of *Artemisia* species.

**Table 1 molecules-27-05128-t001:** Distribution of *Artemisia* L. genus in Central Asian and Russian countries.

Countries	Number of Present Species	Number of Endemic Species
China	186	82
EX-USSR	180	45
Russian Federation	80	-
Kyrgyzstan	55	5
Uzbekistan	47	19
Tajikistan	48	1
Turkmenistan	33	1
Kazakhstan	81	19

(-) not reported.

**Table 4 molecules-27-05128-t004:** Enzymatic activities of *Artemisia* species.

No	Species	α-Glucosidase, Inhibition (%), 50 μg/ml	PTP1B, Inhibition (%), 50 μg/ml	BNA, Inhibition (%), 20 μg/ml
**1**	*A. albida*	55.8	85.5	95.5
**2**	*A. terrae-alba*	25.5	65.2	96.5
**3**	*A. serotina*	25.0	65.0	92.1
**4**	*A. marschalliana*	43.1	75.1	88.1
**5**	*A. schrenkiana*	59.3	76.2	87.2
**6**	*A. rutifolia*	39.3	71.2	85.6
**7**	*A. nitrosa*	47.1	76.1	95.6
**8**	*A. scopaeformis*	83.1	95.6	99.8
**9**	*A. transiliensis*	64.0	92.3	85.6
**10**	*A. scoparia*	28.7	66.5	89.8
**11**	*A. albicerata*	67.8	77.8	95.2
**12**	Deoxynojirimycin *	100.0	-	-
**13**	Ursolic acid *	-	100.0	-
**14**	Quercetin *	-	-	100.0

* Control compounds.

**Table 5 molecules-27-05128-t005:** Antioxidant potentials of *Artemisia* species.

No	Species	TPC mgGAE/100 g	TFC mgQE/100 g	DPPH, IC_50_ (μg/mL)	ABTS, IC_50_ (μg/mL)
1	*A. albida*	832	698	29.9	12.0
2	*A. terrae-alba*	1706	709	14.2	8.8
3	*A. serotina*	1864	748	13.9	9.3
4	*A. marschalliana*	2053	1419	18.0	7.2
5	*A. schrenkiana*	5199	2080	11.7	3.6
6	*A. rutifolia*	3787	2024	10.7	3.7
7	*A. nitrosa*	3856	1892	11.3	2.8
8	*A. scopaeformis*	5804	2745	8.4	1.5
9	*A. transiliensis*	4166	2975	9.4	3.5
10	*A. scoparia*	4711	1951	8.5	3.3
11	*A. albicerata*	3135	1290	13.4	4.9
12	Trolox *	-	-	36.5	19.7

* Positive control.

## Data Availability

Not applicable.

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
