# Peer review of "Traditional Use, Phytochemical Profiles and Pharmacological Properties of Artemisia Genus from Central Asia"

_molecules, 2022, doi:10.3390/molecules27165128_

Round 1

Reviewer 1 Report

This review article provides an overview of Artemisia species from Central Asia, particularly traditional uses in folk medicine and recent numerous phytochemical and pharmacological studies. The study also presented to search promising, potential active Artemisia species candidates, encouraging us to analyze PTP1B, α-glucosidase and BNA inhibition as well as antioxidant potentials of Artemisia plant extracts in which mostly endemic species have not been explored their secondary metabolites and biological activities so far. Before recommending this article for publication, there are some shortcomings for that should be resolve.

General comments

Use accepted complete name of the species.

Species names must be italicized in whole MS.

Abstract

This section is well written but conclusion should be containing one sentence on conclusion and future recommendation.

Also describe about data collection and literature review methods.

Add main results of this study.  

Introduction

Introduction is well written; however, some information could be added to further improve.

Add a general paragraph introduction section on economic, medicinal and historical perspective of plant uses. The following references could be include.

http://dx.doi.org/10.30848/PJB2022-3(19), https://doi.org/10.1016/j.jep.2021.114515,  

Traditional uses, importance of traditional uses at global scale.

Distribution of Artemisia at global scale and regional level.

Discuss main phytochemicals of Artemisia.

Industrial importance and prominent traditional medicines obtained from Artemisia at global and regional level.

Add common uses of oil contents of the species in section 3.

In most of the sentences references are missing.

Add future perspective of this study. Also add potential of the studies species as industrial important plant and facilitating pharmaceutical industries in specific areas.

This manuscript can be accepted after recommended revision.

Author Response

Response to Reviewer 1

This review article provides an overview of Artemisia species from Central Asia, particularly traditional uses in folk medicine and recent numerous phytochemical and pharmacological studies. The study also presented to search promising, potential active Artemisia species candidates, encouraging us to analyze PTP1B, α-glucosidase and BNA inhibition as well as antioxidant potentials of Artemisia plant extracts in which mostly endemic species have not been explored their secondary metabolites and biological activities so far. Before recommending this article for publication, there are some shortcomings for that should be resolve.

→ Thank you for your careful estimation of our manuscript, comments were answered below.

General comments

Use accepted complete name of the species.

→ Accepted complete names of the species were corrected through manuscript.

Species names must be italicized in whole MS.

→ Species names of plant species were italicized through manuscript.

Abstract

This section is well written but conclusion should be containing one sentence on conclusion and future recommendation.

→ In the end of abstract added a sentence on conclusion and recommendation, colored with red.

Also describe about data collection and literature review methods.

→ In abstract added about data collection and literature review methods, colored with red.

Add main results of this study. 

 → In the end of abstract a sentences about main results of this study were corrected, colored with red.

Introduction

Introduction is well written; however, some information could be added to further improve.

Add a general paragraph introduction section on economic, medicinal and historical perspective of plant uses. The following references could be include.

http://dx.doi.org/10.30848/PJB2022-3(19), https://doi.org/10.1016/j.jep.2021.114515,  

A general paragraph introduction section on economic, medicinal and historical perspective of plant uses was added, colored with red. The following references were included.

Traditional uses, importance of traditional uses at global scale.

Traditional uses, importance of traditional uses at global scale were added, colored with red.

Distribution of Artemisia at global scale and regional level.

Distribution of Artemisia at global scale and regional level were added, colored with red.

Discuss main phytochemicals of Artemisia.

Main phytochemicals of Artemisia were discussed, colored with red.

Industrial importance and prominent traditional medicines obtained from Artemisia at global and regional level.

Industrial importance and prominent traditional medicines obtained from Artemisia at global and regional level were added, colored with red.

Add common uses of oil contents of the species in section 3.

Common uses of oil contents of the species were added, colored with red

In most of the sentences references are missing.

 → Missing references were introduced into the text of the article

Add future perspective of this study. Also add potential of the studies species as industrial important plant and facilitating pharmaceutical industries in specific areas.

 Future perspective of this study, potential of the studies species as industrial important plant and facilitating pharmaceutical industries in specific areas were added partially, colored with red.

.

Moreover, the main purpose of this manuscript was reviewing especially Central Asian Artemisia species (for the first time), including their traditional uses, distribution, main phytochemicals, importance and their biological activities. We reviewed all related research papers and books written in different languages, including Kazakh and Russian, because most of the scientific works were collected in Central Asia that were not using English. We sure that for keeping the scientific novelty and exclusivity of work, manuscript introduction and other sections should remain as it is. Although there is too many review papers describing global research situation of Artemisia, because it is one of the most popular plant species for nowadays. We tried to improve introduction in accordance with our purpose, regarding to your comments and added references which you suggested, thanks for understanding.

This manuscript can be accepted after recommended revision.

Thank you for your precise evaluation of our manuscript.

Reviewer 2 Report

The piece of work is interesting and generated some useful information, Overall, the authors summarized the work of Artemisia Genus in the field of traditional use, phytochemical profiles and pharmacological properties, and manuscript was written well to explain the objectives. However, I have some major issues that are to be addressed before the article being accepted. These are as follows:

1. BNA PTP1B,ROS, etc. should be described with full name when first appeared in the text.

2.The MS will be more acceptable if the table 2 can be modified well.

3. It is difficult to follow Figure4-5.What do the series 1 and 2 indicate in these Figures?

4. The tables and figures of the MS should be indicated where they cited.

5. The references are not well prepared and kept uniformity according to the journal instruction. e.g. the journal name in literature 1,2,3,4,etc.The title style of the literature 5,9,etc should be kept uniform.etc.

Author Response

Response to Reviewer 2

The piece of work is interesting and generated some useful information, Overall, the authors summarized the work of Artemisia Genus in the field of traditional use, phytochemical profiles and pharmacological properties, and manuscript was written well to explain the objectives. However, I have some major issues that are to be addressed before the article being accepted.

We are very thankful for your comments, which helped to correct made mistakes.

These are as follows:

  1. BNA,PTP1B, ROS, etc. should be described with full name when first appeared in the text.

Abbreviations were described with full names as it was first appeared in the text.

2.The MS will be more acceptable if the table 2 can be modified well.

Table 2 was improved.

  1. It is difficult to follow Figure4-5. What do the series 1 and 2 indicate in these Figures?

The meanings of series 1 and 2 were corrected in Figures 4-5.

  1. The tables and figures of the MS should be indicated where they cited.

The Tables and Figures indicated to the places where they were cited.

  1. The references are not well prepared and kept uniformity according to the journal instruction. e.g. the journal name in literature 1,2,3,4, etc.The title style of the literature 5,9,etc should be kept uniform.etc.

The references were corrected to kept uniform.